# Ambivalent partnership of the *Drosophila* posterior class Hox protein Abdominal-B with Extradenticle and Homothorax

Jesús R. Curt[1], Paloma Martín[1], David Foronda[1,2], Bruno Hudry[3,4], Ramakrishnan Kannan[5], Srividya Shetty[5], Samir Merabet[3,6], Andrew J. Saurin[3], Yacine Graba[3]*, Ernesto Sánchez- Herrero[1]*

**1** Centro de Biología Molecular Severo Ochoa (CBM), CSIC-UAM, Universidad Autónoma de Madrid, Cantoblanco, Madrid, Spain, **2** Departamento de Medicina, Facultad de Ciencias Biomédicas y de la Salud, Universidad Europea de Madrid, Madrid, Spain, **3** Aix-Marseille Univ., CNRS, Developmental Biology Institute of Marseille (IBDM), UMR 7288, Parc Scientifique de Luminy, Marseille, France, **4** Institut de Biologie Valrose, Université Nice Sophia Antipolis, Faculté des Sciences Parc Valrose, Nice, France, **5** Molecular Genetics lab, Neurobiology Research Center (NRC), National Institute of Mental Health and Neurosciences (NIMHANS), Bangalore, India, **6** Institut de Génétique Fonctionnelle, UMR 5242 CNRS/ENS Lyon, Lyon, France

* yacine.graba@univ-amu.fr (YG); esherrero@cbm.csic.es (ESH)

**Data Availability Statement:** All relevant data are within the manuscript and its Supporting information files.

## Abstract

Hox proteins, a sub-group of the homeodomain (HD) transcription factor family, provide positional information for axial patterning in development and evolution. Hox protein functional specificity is reached, at least in part, through interactions with Pbc (Extradenticle (Exd) in *Drosophila*) and Meis/Prep (Homothorax (Hth) in *Drosophila*) proteins. Most of our current knowledge of Hox protein specificity stems from the study of anterior and central Hox proteins, identifying the molecular and structural bases for Hox/Pbc/Meis-Prep cooperative action. Posterior Hox class proteins, Abdominal-B (Abd-B) in *Drosophila* and Hox9-13 in vertebrates, have been comparatively less studied. They strongly diverge from anterior and central class Hox proteins, with a low degree of HD sequence conservation and the absence of a core canonical Pbc interaction motif. Here we explore how Abd-B function interface with that of Exd/Hth using several developmental contexts, studying mutual expression control, functional dependency and intrinsic protein requirements. Results identify cross-regulatory interactions setting relative expression and activity levels required for proper development. They also reveal organ-specific requirement and a binary functional interplay with Exd and Hth, either antagonistic, as previously reported, or synergistic. This highlights context specific use of Exd/Hth, and a similar context specific use of Abd-B intrinsic protein requirements.

## Author summary

The bilaterians present specific body structures along their antero-posterior axis. This is determined by the activity of Hox proteins, which have been classified in anterior, central

**Funding:** Work in the laboratory of ESH was supported by grants BFU2017-86244-P and PID2020-113318GB-I00 from FEDER/Ministerio de Ciencia e Innovación-Agencia Estatal de Investigación-Consejo Superior de Investigaciones Científicas (https://www.aei.gob.es/). Work in the laboratory of YG was supported by the CNRS (Centre National de la Recherche Scientifique; https://www.cnrs.fr/fr) and AMU (Aix Marseille University https://www.univ-amu.fr/) and grants from AMIDEX n° ANR-11-IDEX-0001-02. Work in the laboratory of RK was supported by DBT-RLF grant by Department of Biotechnology, Government of India. Sanction No BT/RLF/Re-entry /22/20 13 (https://rcb.res.in). The funders had no role in study design, data collection and analysis, decision to publish, or preparation of the manuscript.

**Competing interests:** The authors have declared that no competing interests exist.

and posterior classes, referring to the body region they specify. Hox protein activities are modulated through interaction with partners, that improve DNA binding and regulatory activities. The best-studied partners are Extradenticle (Exd) and Homothorax (Hth) in *Drosophila* (Pbx and Meis/Prep in vertebrates). The study of these partners in the context of the activity of anterior and central Hox proteins demonstrated a broad functional synergism. How Exd and Hth impinge on the activity of Hox proteins specifying the posterior body has been poorly investigated, may be as a result of poorly conserved cofactor interaction motifs in posterior class Hox proteins. We have analyzed the functional interplay of the *Drosophila* posterior class Hox protein Abdominal-B with Exd/Hth. We found an ambivalent partnership ranging from antagonism to synergism relying on molecular requirements that are dependent on the developmental context. Our work starts filling a gap in the understanding of the Exd/Hth-Hox relationship for posterior Hox proteins, already highlighting features not identified when studying anterior and central Hox proteins.

## Introduction

Gene regulation is central for implementing differences between cells of an organism sharing the same genome during development, evolution, and when switches to pathological situations occur. Gene regulation is largely mediated by transcription factors that bind promoters or more distant gene regulatory regions to activate or repress gene expression, the first step towards imprinting differences in protein content within cells. A large number of transcription factors are mobilized for proper gene regulation, in the range of 630 in *Drosophila* to 1500 in human. The large number of transcription factors contrasts with the limited number of strategies evolved to bind DNA, resulting in the grouping of transcription factors in only a few families [1]. The most prominent of these is the zinc finger family, representing almost half of all transcription factors. In this specific instance, large variations in the type and number of zinc fingers allow generating a large repertoire of binding specificity. The second most prominent family of transcription factors is the homeodomain (HD) transcription factor family, with >200 representatives in humans. The HD is a helix-turn-helix DNA binding domain usually 60 amino acid long and highly conserved, in particular within the DNA recognition helix that makes strong contacts with DNA [2]. The high degree of HD conservation endows HD transcription factors with little DNA binding specificity, contrasting with their diverse and specific biological functions [3].

Hox proteins, a sub-group of the HD transcription factor family, have provided a paradigm over years to address the discrepancy between similar DNA binding and specificity in function [4–6]. Hox genes are differentially expressed along the antero-posterior axis, providing positional information for axial patterning in development and evolution [7,8]. A solution to the specificity paradox comes from the observation that Hox proteins act along with partner proteins, often referred to as cofactors, and that the resulting complexes display higher DNA binding specificity. The best studied partners, Pbc and Meis/Prep proteins, are also HD containing proteins, and form dimeric/trimeric DNA binding complexes with increased DNA binding specificity [6,9–14], revealing "latent" specificity of Hox proteins [15–17]. The interactions between Pbc and Hox proteins have been studied by biochemical [18–20] and structural [21–24] methods, and studies in mouse and *Drosophila* have shown that the presence of Exd/Pbx helps to discriminate targets Hox proteins regulate in vivo [17,25,26].

In *Drosophila*, Pbc and Meis/Prep class proteins have a single representative, Extradenticle (Exd) and Homothorax (Hth), respectively, making the genetic analysis easier than in vertebrates (four Pbx, three Meis and two Prep proteins in mouse; [27]). Mutations in *exd* and *hth* modify Hox gene activity without changing Hox expression [28–32]. Early studies established that the formation of a Hox dimeric/trimeric complex was mediated by a short protein motif N-terminal to the HD known as the Hexapeptide (HX) [6,33]. Subsequent studies, however, demonstrated this motif was dispensable for some Pbc-dependent Hox functions [34,35]. Although less extensively documented, sequences just C-terminal to the HD have been shown to mediate interaction with Pbc class proteins. This was shown for the *Drosophila* central class Hox proteins Ultrabithorax (Ubx) and Abdominal-A (Abd-A) [24,36,37]. Structural and molecular modeling identified a short 8 amino acid region, UbdA, only conserved in Ubx and Abd-A, which establishes highly dynamic contacts with Exd [24,36,37]. Interestingly, regions C-terminal to the HD are very often highly conserved within Hox paralogous proteins, suggesting that although the molecular modalities of interaction may vary in between paralog groups, this C-terminal region may generally be dedicated to contribute additional Pbc contacts [38–43]. This was recently established for the vertebrate HoxA1 paralog, for which it was further shown that sequence variation in this protein region is key for functional evolution of Hox1 paralog proteins [44]. The equivalent region in Abd-B is also evolutionary conserved.

Based on the homeodomain protein sequence and expression along the antero-posterior axis, Hox paralogs have been classified as belonging to the anterior, central or posterior groups. In vertebrates, the posterior group comprises Hox genes 9–13 while in *Drosophila* the group includes just one gene, *Abdominal-B* (*Abd-B*). Hox9-10 proteins have a rudimentary HX whose sequence diverges for a core consensual HX motif, yet it displays a central W, and Hox11-13 lack the HX [45]. The lack (or strong degeneration) of the initially identified Pbc interacting motif likely limited the study of how Pbc class (and Meis/Prep classes) functionally interface with posterior class Hox proteins. Most of our current knowledge pertaining to mechanisms of Hox protein specificity thus stems from the study of anterior and central Hox proteins, identifying modes of functional interactions as well as the molecular and structural bases for cooperative action.

*Abd-B* specifies the posterior abdominal segments and some terminal structures in embryos and the posterior abdomen and the genitalia in adults [46–48]. In the *Drosophila* embryo, *Abd-B* represses *exd* and *hth* expression, so that in the last abdominal segment (A8-A9) levels of these two proteins are very low [29–31,49–51]. If *exd* and *hth* expression is maintained in the posterior of the embryo, Abd-B function is impaired, binding of the Hox protein to characterized enhancers is disturbed, and an aberrant phenotype in posterior spiracles is observed [50,51]. These genetic and molecular studies argue for a functional antagonism between the posterior Hox protein Abd-B and Exd/Hth, indicative of a relationship distinct from the one described for anterior and central Hox proteins. However, requirement of *exd/hth* for *Abd-B* to make ectopic embryonic abdominal denticles [52] suggests a more complex interplay. Here we explore how Abd-B function interfaces with that of Exd/Hth using several developmental contexts, studying mutual expression control, functional dependency and intrinsic protein requirements.

## Results

### Lack of Abd-B-mediated repression results in Abd-B, Hth and Exd co-expression in the A7 pupal abdomen

Adult *Drosophila* males have only six abdominal segments as a result of abdominal segment 7 (A7) suppression by Abd-B in pupae [46,47]. In the wild type, the A7 histoblasts divide and the

nests expand as in the rest of abdominal segments during early pupa, but at about 35–42 hours after puparium formation (APF) these cells are extruded and therefore do not form a segment in the adult [53]; by contrast, in *Abd-B* mutants this extrusion does not take place and an A7 is observed [53,54]. As previously described [53–56], there are higher levels of *Abd-B* in A7 than in A6 pupal segments (Fig 1A). Exd and Hth are expressed in all the nuclei of the pupal abdomen including Larval Epidermal Cells (LECs) and histoblasts, and contrary to what happens in the embryo, where Abd-B down-regulates *exd* and *hth* expression [49–51], there is co-expression of Abd-B with Exd and Hth, even in A7, where Abd-B levels are higher (Fig 1B and S1 Fig). This defines a biological context where Abd-B, Exd and Hth are co-expressed, allowing to study how they functionally interface.

To analyze the interaction between *Abd-B* and *exd*/*hth*, we manipulated the expression levels of either *Abd-B* or *exd*/*hth*, and observed the expression of the other. If *Abd-B* expression is reduced in the male A7 by inducing *Abd-B* mutant (*Abd-B^{M5}*) clones, *hth* and *exd* expression does not significantly change (Fig 1C). Analyzed at about 24-30h APF, we found that 36/44 of these clones showed similar Hth or Exd levels to those of cells outside the clones (Fig 1C). However, the remaining 8 clones showed a weak increase of expression (S2 Fig), demonstrating variability in the response to Abd-B loss, which could reflect distinct timing of clonal loss of Abd-B, which in turn may also affect the level of Abd-B protein reduction.

We also used local *Abd-B* inactivation, combining the GAL4/UAS system with the *pannier* (*pnr*)-Gal4 line, active in the central dorsal abdominal region [57], and the UAS-Abd-B RNAi, allowing to compare expression levels in the central (*pnr+*) and lateral (*pnr-*) regions of the same segment and score the *pnr+*/*pnr-* gene levels ratio as a measure of gene regulation, both in experimental and control (*pnr*-Gal4 UAS-GFP) animals. Conditional *pnr*-Gal4 driven expression was achieved using the expression of a thermosensitive form of Gal80 (Gal80^{ts}). In our experimental conditions (see Methods), expression of Abd-B RNAi does not impact, at about 28-32h APF, on Exd and Hth expression in histoblasts in the A7 segment (Fig 1E, compare with Fig 1D, 1H and 1I).

The forced expression of *Abd-B* in the anterior abdominal segments, outside its wild type expression domain, reduces *hth* and *exd* expression: in *Abd-B*-expressing clones in the A3-A4 segments of about 24-28h APF pupae, Exd and Hth expression is strongly reduced in histoblasts while expression in large epidermal cells (LECs) does not seem affected (Fig 1F). Clones in the A6 or A7 segments, where Abd-B levels are high, do not, or only weakly, down-regulate *exd*/*hth* expression (S3 Fig). This suggests that Abd-B levels above high endogenous expression levels do not result in efficient repression. Similarly to the clones, in *pnr*-Gal4 UAS-GFP *tub*-Gal80^{ts} UAS-Abd-B pupae, a reduction of Hth and Exd expression in the *pnr* domain of A3-A5 segments is observed in the histoblasts (but not in LECs), as compared to controls (Fig 1G, 1J and 1K). Thus, while wild type expression of *Abd-B* in A7 does not repress Exd and Hth, its ectopic expression in A3-A4 does.

Collectively, the expression and regulatory interactions studies show there is co-expression of the three proteins in the pupal male A7 segment while in the embryo the repression of *exd* and *hth* by Abd-B prevents the co–expression of the three proteins [49–51]. This difference results from lack of Abd-B-mediated repression on *exd* and *hth* expression in the pupal abdomen, which seems dependent on a precise Abd-B expression levels and/or on different Abd-B regulatory potential at distinct time and location.

## Exd/Hth positively and contextually impacts *Abd-B* expression

Then, we studied if changes in Exd or Hth could affect Abd-B expression. When we induced and analyzed in the male A7 clones mutant for *hth^{P2}*, a null or close to null allele of *hth*, we

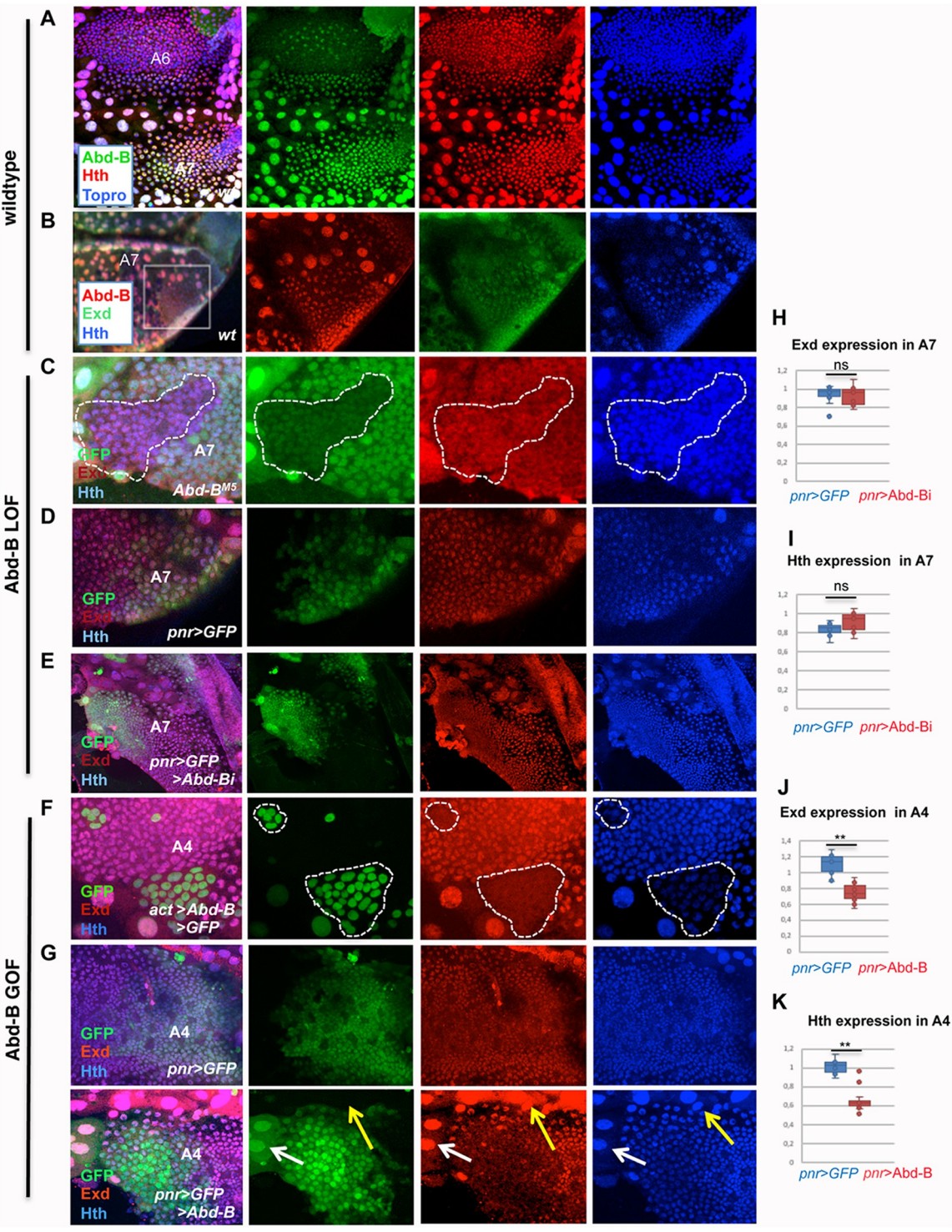

**Fig 1. *Abd-B*, *exd* and *hth* wildtype expression and regulation of *exd* and *hth* by *Abd-B*. (A)** Male wildtype pupa showing higher levels of Abd-B expression (in green) in the A7 than in the A6. Hth is in red and Topro is in blue. **(B)** Male A7 segment of a pupa showing the coincident expression of Abd-B (red), Exd (green) and Hth (blue) in histoblasts and Larval Epidermal Cells. **(C)** *Abd-B^{M5}* clone, induced in the pupal male A7 and marked by the absence of GFP, showing there is no major change in Exd (red) or Hth (blue) expression with respect to wildtype cells. **(D)** In *pnr*-Gal4 UAS-GFP males, the *pnr* domain of the A7 segment has similar levels of *hth* that the *pnr⁻* domain of the same segment. **(E)** The reduction of *Abd-B* expression in the *pnr* region of the male A7 (*pnr*-Gal4 UAS-GFP UAS- Abd-BRNAi) does not change Hth levels of expression with respect to the *pnr⁻* domain. **(F)** The ectopic expression of Abd-B (line 1.1) in the pupal A4 segment, marked with GFP, reduces Exd (red) and Hth (blue) expression with respect to adjacent, wildtype cells.

**(G)** In *pnr*-Gal4 UAS-GFP pupae, the A4 segment has similar levels of expression of Exd (red) and Hth (blue) in the *pnr*⁺ and *pnr*⁻ domains. In *pnr*-Gal4 UAS-GFP UAS-Abd-B (line 1.1) pupae, by contrast, the levels of expression of Exd (red) and Hth (blue) in the *pnr*⁺ domain are substantially reduced in comparison with those of the *pnr*⁻ domain. The white arrow indicate LECs that express GFP (and therefore ectopically express *Abd-B*) but do not reduce Exd or Hth levels as compared with LECs that do not activate Abd-B (yellow arrow). **(H, I)** Quantification of the ratio of expression in a central (*pnr*⁺) with respect to a more lateral (*pnr*⁻) domain of Exd (H) or Hth (I) in *pnr*-Gal4 UAS-GFP/+ and *pnr*-Gal4 UAS-GFP UAS-Abd-BRNAi male A7 histoblast nests. **(J, K)** Quantification of the ratio of expression in a central (*pnr*⁺) with respect to a more lateral (*pnr*⁻) domain of Exd (J) or Hth (K) in *pnr*-Gal4 UAS-GFP/+ and *pnr*-Gal4 UAS-GFP UAS-Abd-B male A3-A4 histoblast nests. Pupae in all panels are of about 24-30h APF. Statistical analysis of the data in H-K was done by two-tailed t-tests, with n = 10–12 pupae.

observed a reduction of *Abd-B* expression in about half of the clones (8/18) (Fig 2A). Similarly, in flip-out clones expressing an *hth RNAi* construct we also found 8/25 clones in the male A7 with reduced *Abd-B* expression (Fig 2B). The clones that reduce *Abd-B* expression are predominantly located in the more posterior region of the segment.

We also analyzed Abd-B expression in the A7 central (*pnr*) and lateral domains of *pnr*-Gal4 UAS-GFP *tub*-Gal80ᵗˢ UAS-exd RNAi, or UAS-hth RNAi, 28-32h APF pupae, and compared the *Abd-B* expression with similarly treated controls. 10 out of 11 pupae of the *pnr*-Gal4 UAS-exd RNAi genotype and 14/15 of the *pnr*-Gal4 UAS-hth RNAi genotype had similar Abd-B levels of expression in both domains (*pnr*⁺ and *pnr*⁻) of this segment, similarly to what is observed in the *pnr*-Gal4/+ controls (Fig 2C, 2E and 2F). The *pnr*-induced loss of Exd or Hth thus seems inefficient in affecting Abd-B expression, suggesting that this mode of down regulation does not sufficiently lower Exd and Hth protein levels. Next, we studied the impact on Abd-B expression of over expression of *hth* in the *pnr* domain. This forced expression does not change the Abd-B levels in A7 with respect to the adjacent, *pnr*-negative domain (Fig 2D and 2G). Flip-out clones expressing *hth* in the A7 do not modify *Abd-B* expression either (Fig 2D). The Hth gain-of-function experiments thus do not show any increase of Abd-B over its wild type levels in A7.

We thus concluded that Exd/Hth positively impact on Abd-B expression, a situation not seen in the embryo [30], and only evidenced by the clonal loss of Exd/Hth. As observed for *exd/hth* regulation by *Abd-B*, the variability observed in *Abd-B* regulation by *exd/hth*, when it does, suggest the need for a precise Abd-B/Hth/Exd levels for proper developmental activity.

## Reduction or increase of *exd/hth* cause *Abd-B* loss of function phenotypes in the male A7

We studied next the role of *exd/hth* in the development of the posterior abdomen, the *Abd-B* domain. *Abd-B* functions in the posterior abdomen by imposing posterior abdominal identity and suppressing the emergence of an A7 segment in the adult male.

Previous analyses showed a transformation of posterior abdominal segments to a more anterior segment in clones or gynandromorphs mutant for *exd* [58]. We also found that reduction of *exd* through *pnr*-mediated RNAi gene inactivation in the central region of the abdomen, or through *hth* loss in *hth^P2* mutant clones, results in the appearance of trichomes, a marker of anterior abdominal segments, in A6, suggesting a transformation of A6 towards a more anterior abdominal identity (Fig 3B–3C', compare with Fig 3A). However, an excess of *hth* in the A6 also causes anteriorwards transformation (ref. [10]; Fig 3D and 3D'). This shows that manipulating *exd/hth* levels in either direction results in an *Abd-B*-like phenotype, thus revealing a complex relationship between *Abd-B* and *exd/hth*, and indicating that *exd/hth* expression levels have to be tightly regulated in the posterior abdomen.

In the wildtype or in heterozygotes for *Abd-B^MD761* there is no A7 (Fig 3E). *Abd-B^MD761* (named as *MD761*-Gal4 in ref. 53) is a P-Gal4 insertion in the *iab-7* regulatory region of *Abd-*

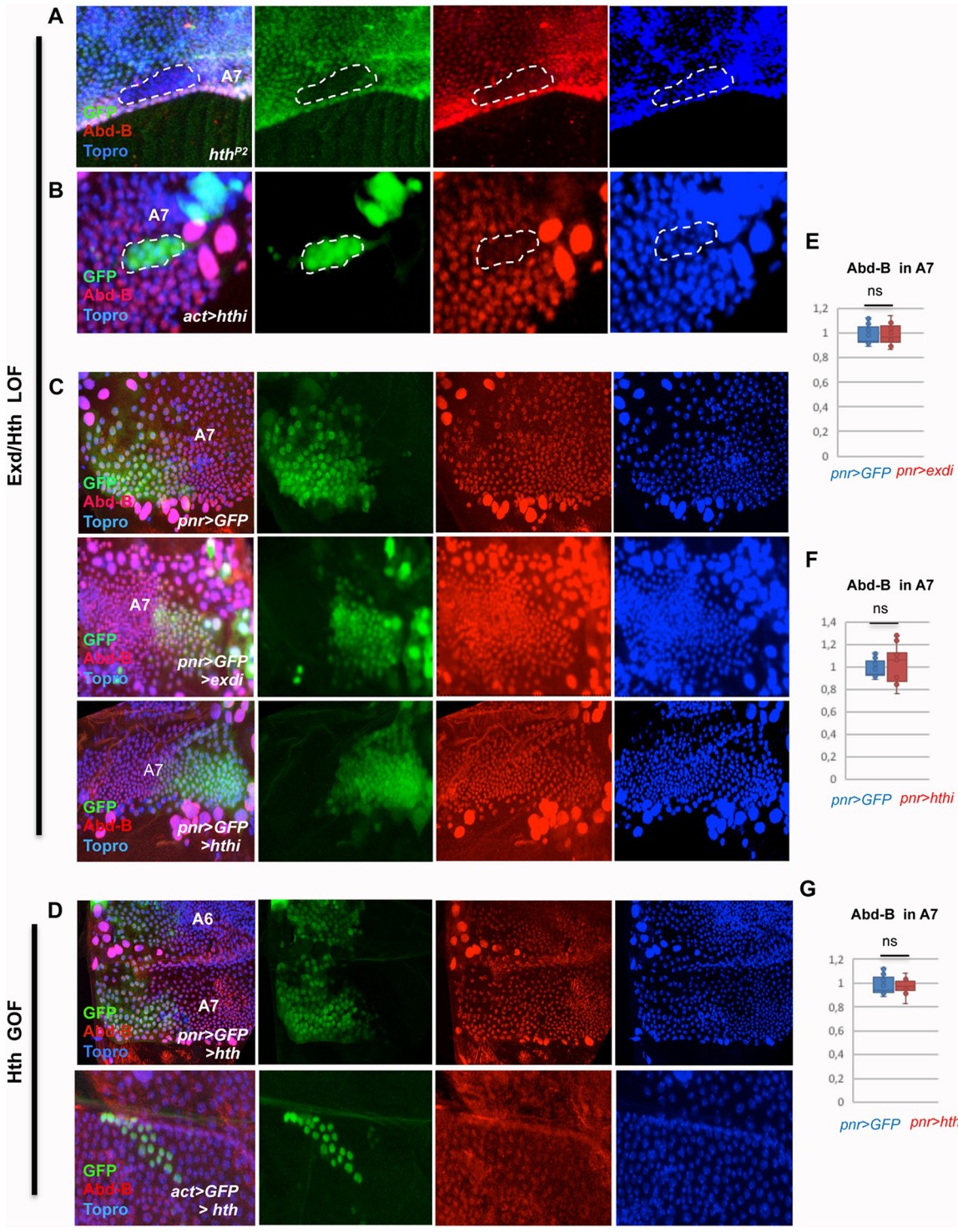

**Fig 2. Regulation of *Abd-B* by *exd* and *hth* in the male pupal abdomen. (A)** *hth^P2* mutant clone in the pupal A7, marked by the absence of GFP, showing reduced expression of Abd-B (red). Topro is in blue. **(B)** Flip-out clone marked with GFP and expressing an hthRNAi construct (act>stop>Gal4 UAS-GFP UAS-hthRNAi) in the male A7 segment showing reduced expression of Abd-B (red). Topro, marking nuclei, is in blue. **(C)** In the male A7 segment of *pnr*-Gal4 UAS-GFP pupae the cells in the *pnr* domain have similar levels of Abd-B (in red) as those not expressing *pnr*. Topro is in blue. If either *exd* or *hth* expression is reduced in the male A7 *pnr* domain (*pnr*-Gal4 UAS-GFP UAS-exdRNAi or *pnr*-Gal4 UAS-GFP UAS-hthRNAi), the expression of Abd-B is not significantly altered with respect to the domain not expressing *pnr*. **(D)** Upper panels: male A7 segment of a *pnr*-Gal4 UAS-*hth* pupa showing similar levels of Abd-B expression (red) in the *pnr*⁺ and *pnr*⁻ domains; lower panels: flip-out clone, marked with GFP, induced in the male A7

segment and expressing *hth*. The levels of Abd-B in the clone (in red) do not change with respect to surrounding cells. **(E-G)** Quantification of the ratio of expression of Abd-B in a central (*pnr*⁺) with respect to a more lateral (*pnr*⁻) domain in *pnr*-Gal4 UAS-GFP/+ and *pnr*-Gal4 UAS-GFP UAS-exdRNAi (E), *pnr*-Gal4 UAS-GFP UAS-hthRNAi (F), and *pnr*-Gal4 UAS-GFP UAS-*hth* (G) male A7 histoblast nests. Pupae in all panels are of about 24-30h APF except in A, which was of about 34h APF. Statistical analysis of the data in E-G was done by two-tailed t-tests, with n = 10–12 pupae.

*B* that drives expression in the A7 segment (histoblasts and LECs) and is also mutant for the *iab-7* function of *Abd-B*, required for A7 suppression [53]. Loss of this suppression (emergence of an A7) is obtained when combining *Abd-B*^MD761^ with the *Abd-B*^M1^ mutation (Fig 3F). The size of the A7 segment can be assessed by the number of bristles in the A7 segment, providing an accurate estimate of the segment size, and therefore of Abd-B activity (Fig 3R). A reduction of *exd* or *hth* by driving *exd* or *hth* RNAi expression with the *Abd-B*^MD761^ driver produces a segment in the A7 position that also bears anterior identity, phenocopying (although not as strongly) the lack of *Abd-B* (Fig 3G and 3H and S4 Fig for simultaneous reduction of *exd* and *hth*). As observed in A6 (Fig 3D and 3D'), increased expression of *exd*, and to a lesser extent of *hth*, also results in the appearance of A7 segments (Fig 3J and 3K). Combined expression of both *exd* and *hth* results in a phenotype very similar to that of *exd* expression only (S4 Fig).

These results suggest a genetic interaction between *exd*/*hth* and *Abd-B* in male A7 development. To assess further this relationship, we explored the dependency of the *exd*/*hth*-induced phenotypes upon *Abd-B* expression levels. We observed that the A7 that develops when there is a reduction of *exd* is completely suppressed by increasing the doses of *Abd-B* from 2 to 3 with the introduction of DpP5, a duplication for the three genes of the Bithorax Complex ([59]; compare Fig 3H with Fig 3I and 3S). This is consistent with the *Abd-B* down-regulation observed in some *hth*^P2^ mutant clones (Fig 2A). While an excess of *exd* or *hth* does not reduce *Abd-B* expression (Fig 2D), an *Abd-B* loss of function phenotype is also observed when *exd* expression is increased in the A7 (Fig 3K), suggesting a reduction of Abd-B activity. Consistently, the concomitant expression of *Abd-B* and *exd* reduces the A7 formed by an excess of *exd* (compare Fig 3K with Fig 3L and 3T). These results suggest two different mechanisms by which excess or reduction in *exd*/*hth* levels regulate Abd-B function, one by controlling its level of expression and the other by modulating its activity.

We also assessed further the relationship between the *Abd-B* and *exd*/*hth* phenotypes by exploring dependency of the *Abd-B*-induced phenotypes upon Exd levels. To this end, we compared the rescuing effect of *Abd-B* in the development of male A7 in the presence and in the absence of *exd*. As described above, the *Abd-B*^MD761^/*Abd-B*^M1^ combination results in the development of an A7 in the male (Fig 3F). The expression of *Abd-B* in this mutant background (taking advantage of *Abd-B*^MD761^ being a Gal4 line) partially rescued this transformation (compare Fig 3F and Fig 3M; the Gal4/Gal80^ts^ system was used in these rescuing experiments, see Methods). If *exd* expression is simultaneously reduced in this "rescuing" background, A7 development is not significantly altered (Fig 3M, 3N and 3U). This result in isolation would suggest that *Abd-B* does not need *exd* for A7 development. However, the findings reported in Fig 3F–3H argue for a common role of Abd-B and Exd/Hth in A7 suppression and identity. In addition, the finding that both the loss and gain of Exd and Hth results in similar A7 phenotypes, which underlines the importance of relative Abd-B and Exd/Hth levels, rather suggest that in the rescuing conditions used, we do not modify sufficiently the Abd-B to Exd/Hth ratio to impact on the remaining Abd-B A7 suppressive function. Supporting the importance of this ratio, if higher levels of *Abd-B* (obtained using the UAS-Abd-B line 1.1; [60]) are induced in the *Abd-B*^MD761^/*Abd-B*^M1^ background, the mutant phenotype is completely rescued, and the effect is independent of the presence of *hth* (Fig 3O–3Q).

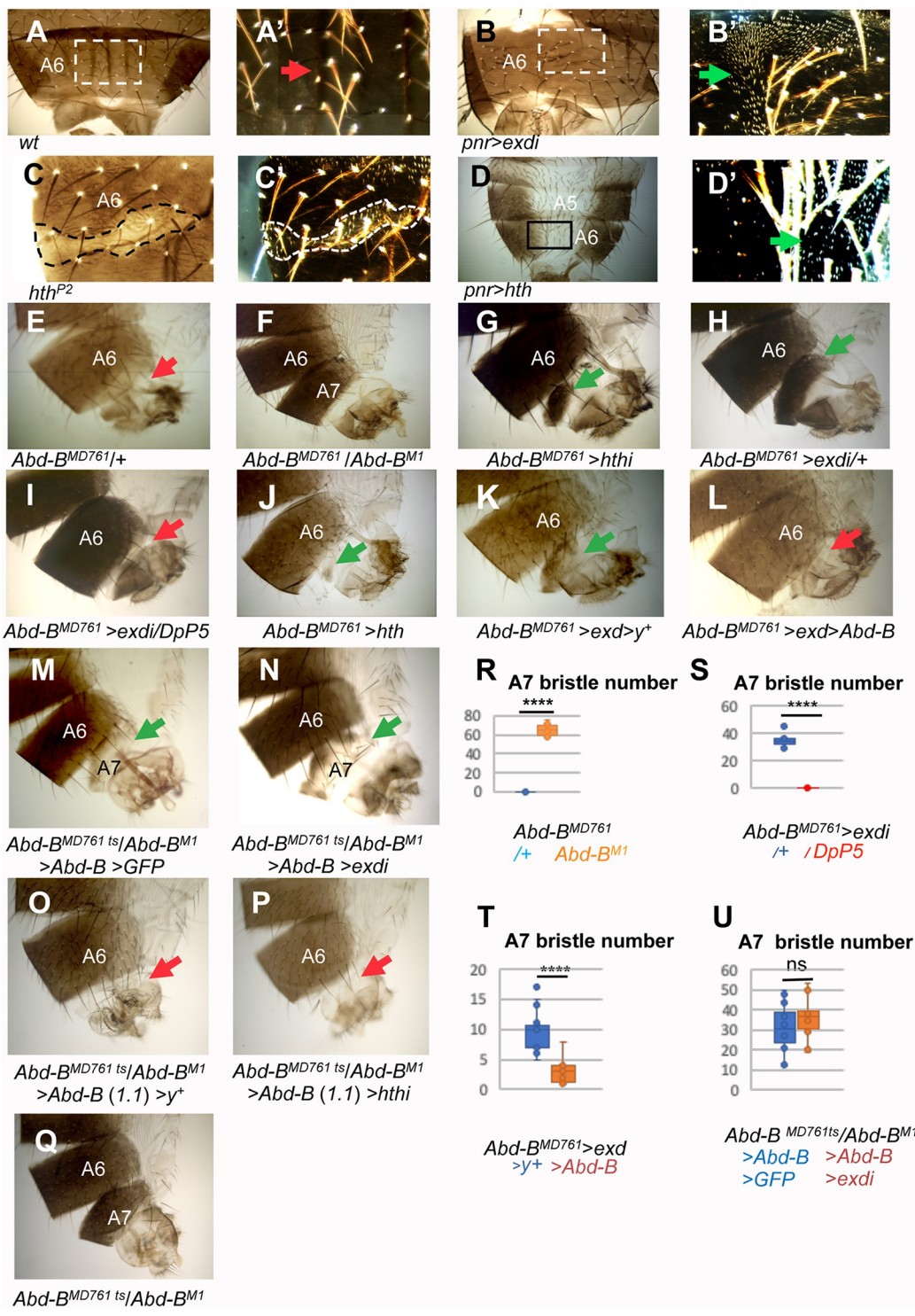

**Fig 3. Reduction or excess of *hth/exd* cause anteriorwards transformations in the male abdomen. (A, A')** Bright (A) and dark field (A', inset) images of the dorsal A6 segment of a wildtype male, showing no trichomes in the medial region of the segment (red arrow). **(B, B')** In *pnr*-Gal4 UAS-exdRNAi males, the dorsal central domain of the A6 segment, where *pnr* is expressed, presents many trichomes (B', dark field image of inset, green arrow), suggesting transformation to a more anterior segment. **(C, C')** Bright (C) and dark field (C') images of a clone mutant for *hth^P2^*, marked with *yellow* and outlined, showing trichomes within the clone, suggesting transformation to a more anterior segment. **(D, D')** In the *pnr*-Gal4 UAS-*hth* abdomen, the central dorsal region of A5 and A6, where *pnr* is expressed, present depigmentation and a great number of trichomes (inset of D in D', dark field, green arrow), again suggesting transformation to a more anterior

segment. **(E)** $Abd\text{-}B^{MD761}$/+ males have no A7 segment, like the wildtype. In this and subsequent panels the red arrows indicate absence of A7 and the green arrows presence of this segment. **(F)** $Abd\text{-}B^{MD761}$/$Abd\text{-}B^{M1}$ male (mutant background for rescue experiments). The A7 is largely transformed into the A6. **(G-I)** The reduction of either *hth* (G) or *exd* (H) results in the development of a small A7 segment. The formation of this segment depends on the amount of Abd-B, since an extra dose of *Abd-B* (with DpP5) in an $Abd\text{-}B^{MD761}$ UAS-exdRNAi background (I) reverts the mutant phenotype to the wildtype (compare H with I). **(J-L)** The increase in the amount of *hth* (J) or *exd* (K) also forms an A7 segment, and this development is prevented if *Abd-B* is concomitantly expressed (L). **(M, N)** The expression of *Abd-B* in a *tub*-Gal80$^{ts}$/UAS-*Abd-B*; $Abd\text{-}B^{MD761}$ UAS-GFP/$Abd\text{-}B^{M1}$ male (shift from 18°C to 29°C at third larval stage) partially reverts the phenotype of the $Abd\text{-}B$ mutant background (M, compare with F and Q); this phenotype does not significantly change if there is reduction of *exd* (UAS-exdRNAi/UAS-*Abd-B*; $Abd\text{-}B^{MD761}$ *tub*-Gal80$^{ts}$/$Abd\text{-}B^{M1}$ male with the same temperature shift) (N). **(O)** UAS-$y^+$/ UAS-*Abd-B (1.1)*; $Abd\text{-}B^{MD761}$ *tub*-Gal80$^{ts}$/$Abd\text{-}B^{M1}$ male (shift from 18°C to 29°C at the third instar) showing complete suppression of A7 development. **(P)** The reduction of *hth* expression in UAS-*Abd-B (1.1)*/+; $Abd\text{-}B^{MD761}$ *tub*-Gal80$^{ts}$/$Abd\text{-}B^{M1}$ UAS-hthRNAi males, with a similar temperature shift, shows also no A7 segment. **(Q)** $Abd\text{-}B^{MD761}$ *tub*-Gal80$^{ts}$/$Abd\text{-}B^{M1}$ male (the mutant background for experiments in O, P), also shifted from 18°C to 29°C in third instar larva, showing transformation of A7 into A6. **(R-U)** Quantification of the number of bristles in the following genotypes: $Abd\text{-}B^{MD761}$/+ (n = 11) and $Abd\text{-}B^{MD761}$ /$Abd\text{-}B^{M1}$ (n = 14) (**R**), $Abd\text{-}B^{MD761}$ UAS-exdRNAi, and $Abd\text{-}B^{MD761}$ UAS-exdRNAi DpP5 (n = 10 for both genotypes) (**S**), $Abd\text{-}B^{MD761}$ UAS-*exd* UAS-$y^+$ (n = 14) and $Abd\text{-}B^{MD761}$ UAS-exd UAS-*Abd-B* (n = 10) (**T**), and *tub*-Gal80$^{ts}$/UAS-*Abd-B*; $Abd\text{-}B^{MD761}$ UAS-GFP/$Abd\text{-}B^{M1}$ (n = 18) and UAS-exdRNAi/UAS-*Abd-B*; $Abd\text{-}B^{MD761}$ *tub*-Gal80$^{ts}$ /$Abd\text{-}B^{M1}$ males (n = 14) (**U**). Statistical analysis of the data in R-U was done by two-tailed t-tests.

The importance of the Abd-B to Exd/Hth ratio is further supported by the analysis of *wing-less* (*wg*), an Abd-B target in the abdomen. The expression of *wg* in the male A7 is suppressed by *Abd-B* ([53,54]; Fig 4A). If *exd* levels are reduced in this segment, ectopic *wg* expression is also observed (Fig 4A). We have found, however, Wg antibody expression in the male A7 sometimes difficult to observe precisely due to high background and to the curvature of the posterior abdomen, so we turned to analyze Abd-B activity by observing the expression of a Wg-GFP protein trap [61] and in the A3-A4 abdominal segments. We found that by increasing *Abd-B* levels in the central dorsal abdomen with *pnr*-Gal4, Wg-GFP expression in the male A3-A4 segments is strongly reduced (Fig 4B and 4C). Increasing Exd levels in this context impairs Abd-B ability to down-regulate Wg-GFP expression, suggesting high levels of Exd counteract the repressing activity of Abd-B (Fig 4B and 4C).

Collectively, these data set indicate that Abd-B and Exd/Hth function converge in controlling A7 development, including A7 suppression and posterior identity specification, and that accurate levels of both components are required for proper Abd-B activity and A7 development.

## Abd-B associates with Exd/Hth in the pupal abdomen independently of the generic HX interaction motif

Excess of *exd* or *hth* results in partial loss of A7 suppression, producing an *Abd-B* mutant-like phenotype without lowering *Abd-B* expression levels. This suggests that Exd/Hth reduces Abd-B activity, which could be possibly mediated through physical inhibitory interaction between Exd/Hth and Abd-B proteins, as previously observed [50,51]

We first probed the possibility of physical interactions between Abd-B and Exd/Hth by co-immunoprecipitation after producing the proteins in *Spodoptera frugiperda* cells (*SF*9 baculovirus expression system; [62]). We infected SF9 cells with baculovirus expressing tagged Exd, Hth or Abd-B proteins (His::Exd, Flag::Hth and HA::Abd-B) and incubated SF9 extracts on a resin coupled to anti-Flag. After elution we observed co-immunoprecipitation of Abd-B and Exd (Fig 5A), whereas if Flag::Hth was omitted, neither Abd-B nor Exd was bound by the resin. In the absence of Exd, Hth still co-precipitates with Abd-B, suggesting direct Abd-B/Hth interactions. The existence of this dimeric complex may explain that the gain of function of Hth results in the reduction of A7 suppression: if the A7 suppressive complex is Abd-B/Exd/

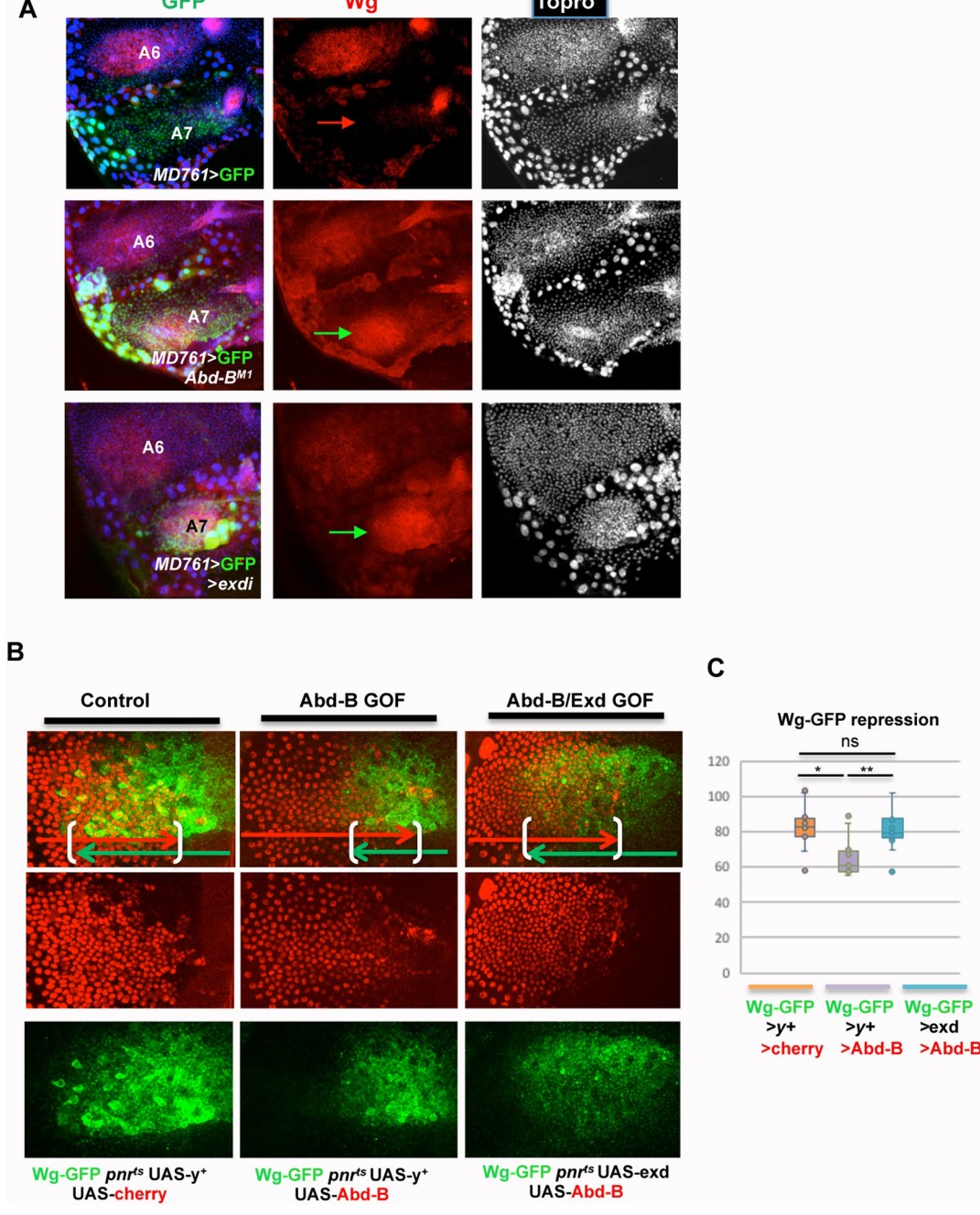

**Fig 4. Regulation of wg expression by *exd/hth* and *Abd-B*.** (A) Expression of Wg in the male A7 of *Abd-B^{MD761}* UAS-*GFP*/+, *Abd-B^{MD761}* /*Abd-B^{M1}*, and *Abd-B^{MD761}*/UAS-exdRNAi males, showing no Wg expression in the A7 in the first genotype (red arrow) and ectopic Wg expression in this segment of the two latter genotypes (green arrows). (B) Expression of wg-GFP in the A4 segment of male pupae: when *Abd-B* is overexpressed (wg-GFP/UAS-*Abd-B*; *pnr*-Gal4 *tub*-Gal80^{ts}/UAS-*y^+*; n = 10) there is partial repression of wg-GFP, as compared to the control (wg-GFP/UAS-*y^+*; *pnr*-Gal4 *tub*-Gal80^{ts}/UAS-*cherry*; n = 10). This repression is partially reverted by the concomitant expression of *exd* (UAS-*exd*; wg-GFP/UAS-*Abd-B*; *pnr*-Gal4 *tub*-Gal80^{ts}/+; n = 12). Larvae of the three genotypes were shifted from 18°C to 29°C in the third larval stage. (C) Quantification of the extent of overlap of the wg-GFP and cherry, or wg-GFP and Abd-B, signals in the three genotypes. Statistical analysis was done by One-way ANOVA.

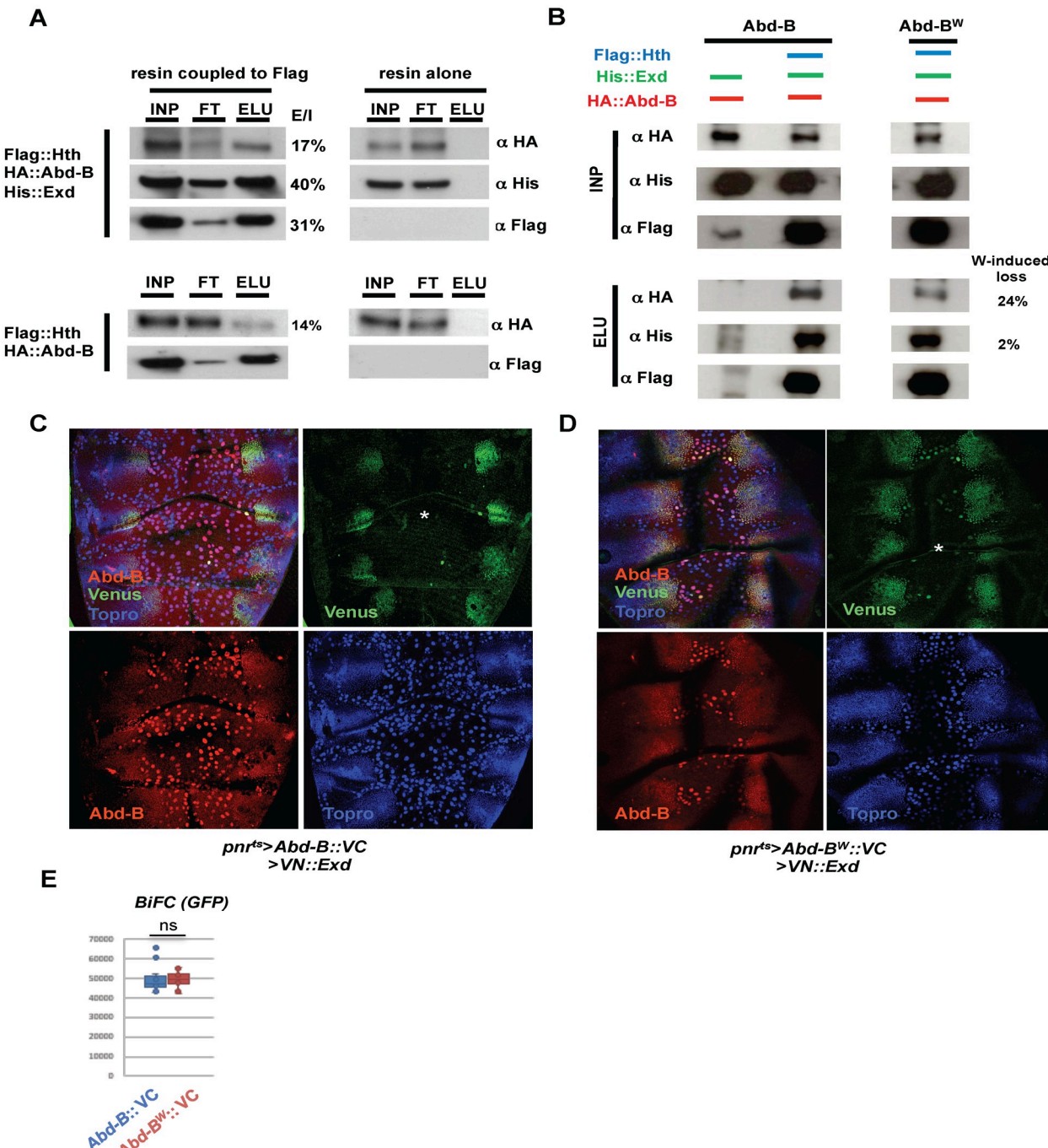

**Fig 5. Co-immunoprecipitation and BiFC experiments related to Abd-B and Exd interaction. (A)** Abd-B, Exd and Hth co-immunoprecipitations (co-IPs). INP, FT, and Elu stand, respectively, for input (the material loaded on the resin), flow through (the material not captured by the resin), and elution (the material bound to the resin and boiled eluted). Protein present in the INP, FT and ELU is detected by the fused tags: Flag for Hth, which also serves for capturing on the anti-Flag resin, HA for Abd-B and His for Exd. The resin not coupled to anti-Flag serves as a control (three right columns). Above is the co-IP in the presence of Hth, Abd-B and Exd, and below the co-IP in the presence of Hth and Abd-B. Abd-B co-precipitates with Hth in the presence and absence (to a lesser extent) of Exd. % next to the panels indicates the ratio Elu/INP (x100). **(B)** Comparative Abd-B and Abd-B^W co-IPs. Protein present in the co-IPs is indicated above the gels. There is a significant decrease in the amount of the Abd-B^W protein captured on the resin when compared to the Abd-B wildtype protein (24%), while Exd remains almost unchanged (2%). **(C, D)** Dorsal views of pupae of approximately 24-28h APF of the genotypes *pnr*-Gal4 *tub*-Gal80^ts UAS-*Abd-B*::*VC* UAS-*VN*::*Exd* (C) and *pnr*-Gal4 *tub*-Gal80^ts UAS-*Abd-B*^W::*VC* UAS-*VN*::*Exd* (D) showing Venus signal (in green), Abd-B expression (in red) and Topro, marking nuclei (in blue). See that there is Venus signal, indicating complementation between the Venus fragments of the Abd-B and Exd proteins in histoblasts but not (or just a few) in the polytene LECs (asterisks). The preparation in D has a reduced number of LECs due to tearing of the tissue during

mounting. **(E)** Quantification of BiFC signals. There is similar Venus signal in the experiments performed with the Abd-B and Abd-B$^W$ constructs (n = 12 for the two genotypes). Statistical analysis was done by two-tailed t-test.

Hth, increased Hth expression may favor the formation of a dimeric non-functional Abd-B/Hth complex that competes with the activity of the A7 suppressive Abd-B/Exd/Hth one. The same rational may as well explain the similar effects of Exd loss and gain of function.

The interaction between Exd and Hox proteins is generally mediated by the Hexapeptide (HX), found in most Hox proteins shortly before the HD [33]. In the Abd-B protein the HX diverges from the canonical core HX sequence, but a Tryptophan that defines the core of the HX motif is present [6,63]. To probe the involvement of the Abd-B HX-like motif in Exd/Hth interactions, we repeated the in vitro co-immunoprecipitation experiments using an Abd-B protein bearing a mutation of this Tryptophan towards an Alanine (Abd-B$^W$). In these experiments, we observed a reduction in the amount of co-precipitating Abd-B$^W$ (24%), compared to wild type Abd-B (Fig 5B). The Abd-B$^W$ protein that remains bound to the Hth-resin could result from Abd-B-Exd interaction not dependent upon the HX like motif, and/or from direct interaction of Abd-B with Hth, as seen in the experiments conducted with the wild type Abd-B protein (Fig 5A). Together, these data demonstrate that Abd-B associates with Exd/Hth by binding to both proteins, and that the generic HX interaction motif contributes, but is not essential, for these interactions.

We next studied protein interactions directly in the posterior abdomen by Bimolecular Fluorescence Complementation (BiFC) using the Gal4/UAS system [64]. We used UAS constructs in which the coding regions for the N-terminal or C-terminal parts of the Venus fluorescent protein are fused to the coding regions of the Exd or Abd-B proteins (UAS-*VN*::*Exd* and UAS-*Abd-B*::VC) [39,64]. We expressed simultaneously the two constructs in the dorsal abdomen of third instar larvae and pupae with the *pnr*-Gal4 line and the Gal4/Gal80$^{ts}$ system. We observed BiFC in the region of co-expression, the dorsal central abdomen (Fig 5C), including the posterior segments where endogenous Abd-B is expressed. Signal resulting from BiFC is almost exclusively confined to histoblasts and not present in LECs, despite expression of the *VN*::*Exd* and *Abd-B*::*VC* proteins in both cell types. This result indicates that Abd-B associates with Exd in the pupal abdomen, specifically in histoblasts and not in LECs. To probe for the contribution of the HX-like motif, BiFC experiments were repeated using a UAS-*Abd-B$^W$*::*VC* construct inserted at the same chromosomal position, allowing for similar expression levels. Results showed a level of BiFC very similar to that seen with the wild type Abd-B::VC protein (Fig 5D and 5E). We concluded that in the posterior abdomen, Abd-B associates with Exd, specifically in histoblasts, and that the core Tryptophan of the HX-like motif does not contribute to this interaction.

## Abd-B protein requirement of A7 suppression: Dispensability of the HX central Tryptophan residue but requirement of the HX like motif and posterior-Hox specific protein sequences

While the SF9 co-IP experiments identify a contribution of the HX-like motif to Exd interaction, the BiFC data in the A7 abdomen indicates that it is not necessary for interaction with Exd, suggesting it may be dispensable for A7 suppression by *Abd-B*. To probe this, we used, as before, a rescuing experiment combining *Abd-B$^{MD761}$*/*Abd-B$^{M1}$*, which alleviates the ability of *Abd-B* to suppress A7 (Fig 3F), with UAS lines allowing for Abd-B or Abd-B$^W$ expression (Fig 6A shows a scheme of the rescuing experiments). In the condition of our experiments (see Methods), the expression of the Abd-B wildtype protein partially corrects the Abd-B mutant

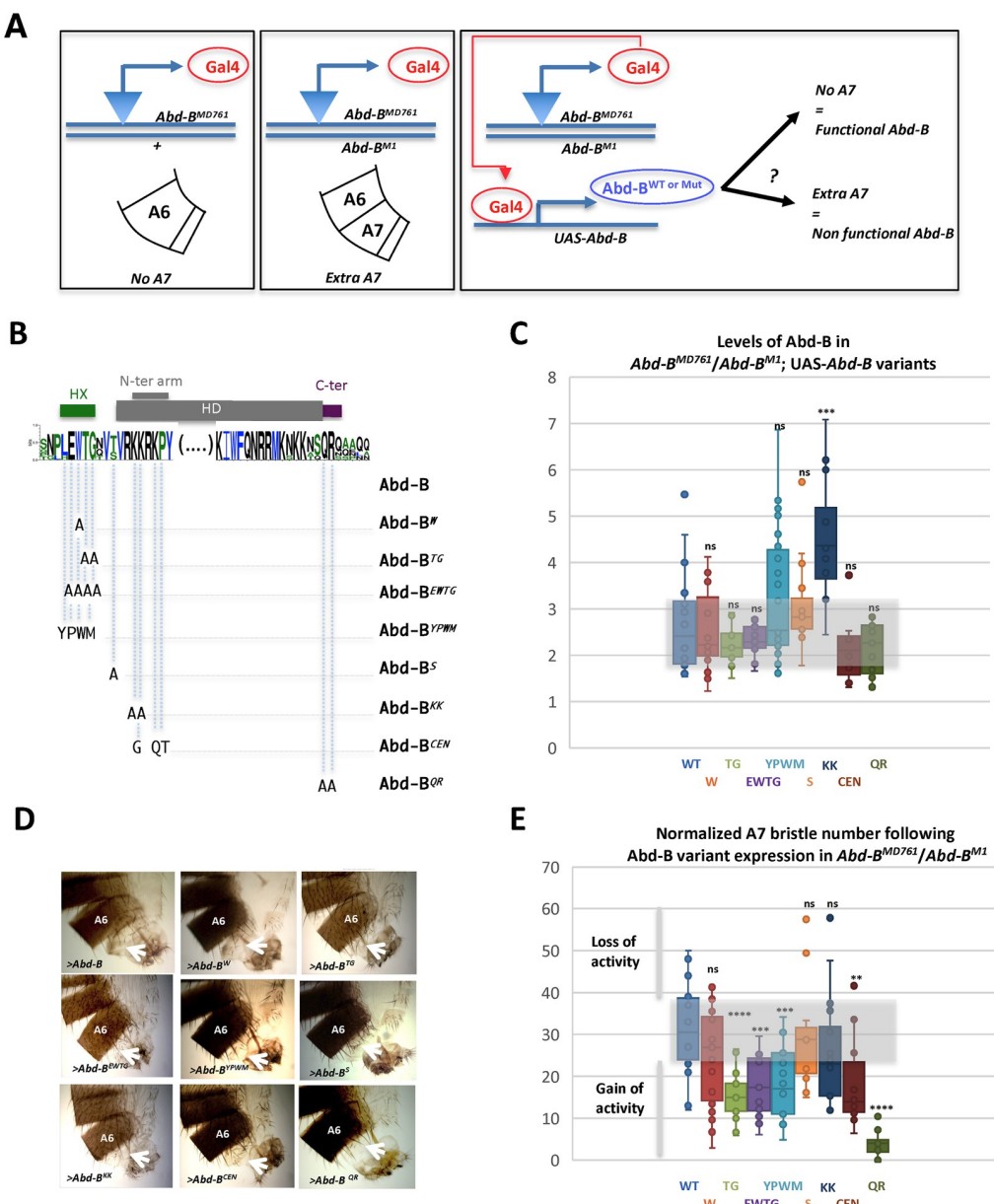

**Fig 6. Activity of the wildtype and Abd-B$^W$ mutant proteins. (A)** Scheme of the genetic experiments performed to ascertain the rescue of the male *Abd-B$^{MD761}$/Abd-B$^{M1}$* mutant phenotype (A7 transformed into A6) by the expression of wildtype *Abd-B* or different *Abd-B* variants. **(B)** Scheme of Abd-B protein mutations. The sequence of the Abd-B protein is indicated, as well as its most conserved domains (in arthropods and human), the Homeodomain (HD), the Hexapeptide (HX), and sequences immediately C-terminal to the HD. The N-terminal arm of the HD is highlighted. Web logo was obtained using sequences for the following Abd-B sequences (Drosophila (AAA84402), Tribolium (AAF36721.1), Anophela (XM311628), Sacculine (AAQ49317.1), Folsomia (AAK52499.1) and human (BCO10023). **(C)** Levels of Abd-B protein expression in the rescue experiment. Levels of the wild type (WT) and different Abd-B variants (referred to according to the labeling in B) measured as amount of protein in a defined area of A7/amount of protein in the same area of A6 dorsal histoblast nests in 24-28h APF pupae; n = 20 (Abd-B), 19 (W), 11 (TG), 11 (EWTG), 19 (YPWM), 15 (S), 12 (KK), 11 (CEN) and 15 (QR). **(D)** Examples of the posterior abdomens of males expressing wild type or Abd-B variants (UAS constructs represented as > Abd-B [x]) in the *Abd-B$^{MD761}$/Abd-B$^{M1}$* common genetic background. Arrows point to the A7. **(E)** Quantification (measured as number of bristles) of the rescue of the A7 mutant phenotype observed in *Abd-B$^{MD761}$/Abd-B$^{M1}$* animals by the different Abd-B protein variants. Bristle numbers were normalized relative to the level of Abd-B and Abd-B protein variant expression (counts x ratio Abd-B variants/ Abd-B); n = 18 (Abd-B), 26 (W), 15 (TG), 16 (EWTG), 21 (YPWM), 11 (S), 16 (KK), 14 (CEN) and 12 (QR). Statistical analysis on normalized bristle number values was performed using the Wilcoxon test.

phenotype, with an average of 31,6 bristles, while the $Abd\text{-}B^{MD761}/Abd\text{-}B^{M1}$ A7 segments display an average of 65 bristles (Figs 3F, 6D and 6E). With the expression levels of wild type and Abd-B$^W$ proteins being very similar (Fig 6C), scoring of A7 bristles indicates that the rescuing ability of A7 suppression by both proteins is also very similar (Fig 6D and 6E). This result supports that the Tryptophan residue of the HX-like motif is neither required for Exd interaction in A7 (Fig 5C–5E), nor for Abd-B A7 suppressive activity.

Facing the lack of requirement of the W residue for Exd-dependent Abd-B function in male A7 suppression, we investigated more broadly possible contributions of other Abd-B protein sequences (51; Fig 6B): mutations that target amino acids located within the EWTG HX like motif (W>A, TG>AA, EWTG>AAAA or EWTG>YPWM, the sequence of a canonical HX motif), within the short linker region connecting the HX to the HD (an evolutionarily conserved S, S>A), within the HD N-terminal arm known to provide paralog specificity [15,16], either mutating the posterior class specific signature (KK>AA) or switching it towards a central class specific signature (CEN: KRKP> GRTQ), or within sequences just downstream to the HD (Cter: QR>AA), shown to provide an alternative Exd interaction motif in the central Ubx and Abd-A Hox proteins [36–38]. Sequences mutated are evolutionary conserved in Abd-B proteins from other insects.

Before probing the functional impact of Abd-B mutations, we studied the expression levels of all Abd-B variants in the abdominal A7 (Fig 6C). Taking the expression level of the $Abd\text{-}B^{MD761}$-driven wild type protein as a reference point, we found that most proteins are expressed within a similar range of expression level. The most significant exception is the Abd-B$^{KK}$ protein, whose expression is significantly higher than that of the wild type Abd-B protein. Differences in expression levels (subtle as well as stronger differences seen for Abd-B$^{KK}$) were taken into account to normalize the rescuing ability of Abd-B variants (see Methods). Results show that none of the mutations results in a weakening of Abd-B activity (Fig 6D and 6E). On the contrary, several mutations lead to increased Abd-B A7 suppressive activity. This includes mutations within the HX like motifs (in particular Abd-B$^{TG}$, Abd-B$^{EWTG}$, and Abd-B$^{YPWM}$), within the HD N-terminal arm (Abd-B$^{CEN}$) and in the conserved sequence C-ter to the HD (Abd-B$^{QR}$).

Thus, while the core Tryptophan residue within the HX motif is dispensable for A7 suppression, other residues within the Abd-B HX-like motif are required, as are protein sequences specific to posterior Hox class proteins, in positions known to mediate interaction with Exd in other Hox proteins.

## Context specific usage of intrinsic protein determinants for Abd-B functions

To further probe for a possible function of the W residue of the Abd-B HX like motif, we examined its requirement for development of the female genitalia, which include easily scored structures, the vaginal teeth, which form two rows easily identifiable (Fig 7A). Vaginal teeth are influenced by *Abd-B* (reduction of *Abd-B* expression eliminates female genitalia, including vaginal teeth, ref. 65; illustrated in Fig 7B for the $Abd\text{-}B^{LDN}/Abd\text{-}B^{M1}$ combination), *hth* ($hth^{P2}$ mutant clones result in more and disorganized vaginal teeth, ref. 66), and *exd* (reduction of *exd* produce more, bigger and disorganized vaginal teeth, Fig 7C).

To analyze the function of the different Abd-B variants in female genitalia development we used a rescue experiment similar to the one described for probing restoration of A7 suppression in the male. Instead of the $Abd\text{-}B^{MD761}$ line, we used the $Abd\text{-}B^{LDN}$ line that is mutant for *Abd-B* female genitalia function, and drives expression of Gal4 in female genitalia, mimicking Abd-B expression (65; see scheme in Fig 7D). In $Abd\text{-}B^{LDN}/Abd\text{-}B^{M1}$ females the genital disc is

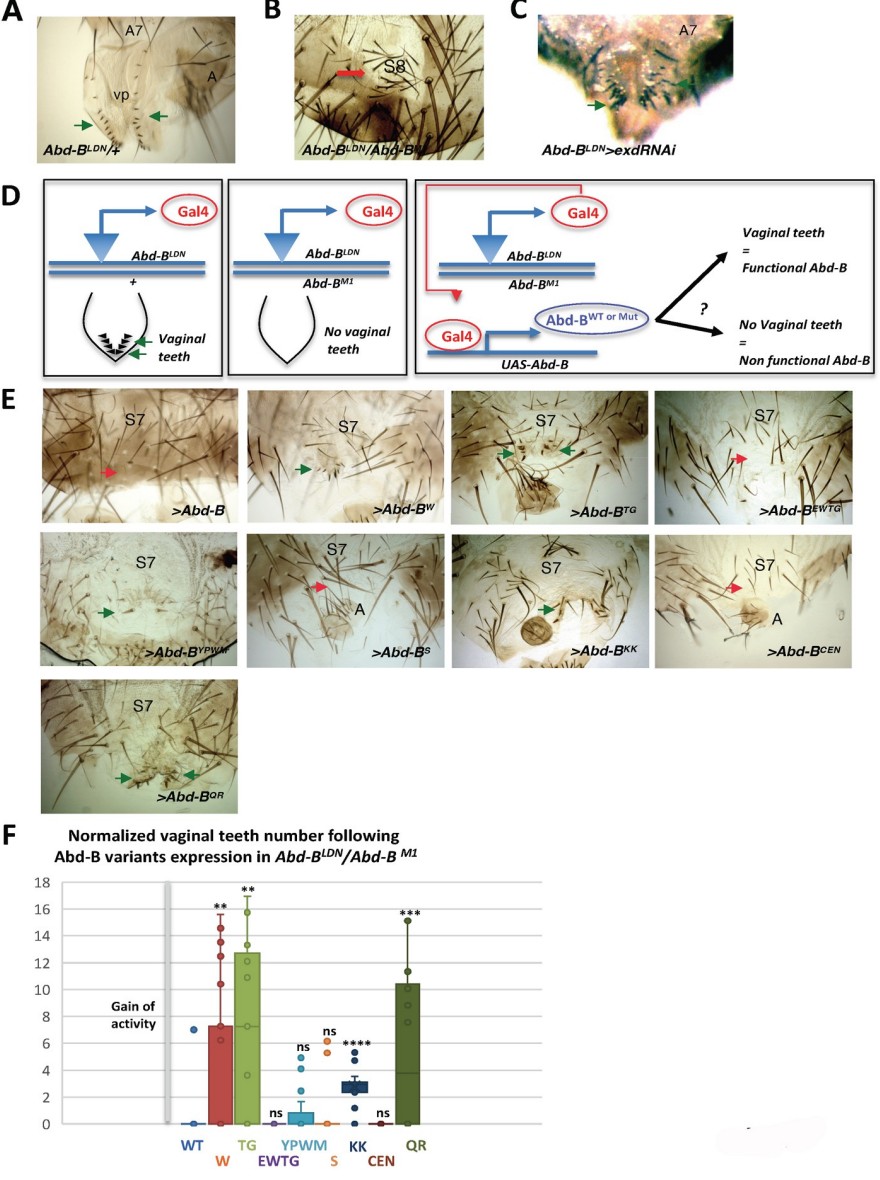

**Fig 7. Rescue of the female genitalia mutant phenotype by different Abd-B proteins. (A)** *Abd-B*<sup>LDN</sup>/+ female genitalia, showing the two rows of vaginal teeth (arrows). Vp, vaginal plates. **(B)** In *Abd-B*<sup>LDN</sup>/*Abd-B*<sup>M1</sup> females the genitalia are eliminated and frequently replaced by an eighth sternite (S8). **(C)** Reduction of *exd* produces more, bigger and disorganized vaginal teeth (arrows). **(D)** Scheme of the genetic experiments performed to ascertain the rescue of the *Abd-B* mutant phenotype of *Abd-B*<sup>LDN</sup>/*Abd-B*<sup>M1</sup> females (elimination of genitalia, including vaginal teeth) by the expression of wildtype or different *Abd-B* variants. **(E)** Examples of female genitalia phenotypes of *Abd-B*<sup>LDN</sup>/*Abd-B*<sup>M1</sup> (common genetic background) expressing wild type or Abd-B variants (UAS constructs represented as > Abd-B [x]), showing no rescue of the mutant phenotype (red arrows) or a small rescue, with presence of some vaginal teeth (green arrows). S7, seventh sternite; A, analia. **(F)** Quantification (measured as number of vaginal teeth) of the rescue of the absence of genitalia phenotype observed in *Abd-B*<sup>LDN</sup>/*Abd-B*<sup>M1</sup> animals expressing the *Abd-B* wild type or the Abd-B variants; n = 24 (Abd-B), 28 (W), 11 (TG), 12 (EWTG), 19 (YPWM), 17 (S), 12 (KK), 13 (CEN) and 20 (QR). Vaginal teeth numbers were normalized relative to the level of Abd-B and Abd-B protein variant expression (counts / ratio Abd-B variants/ Abd-B).

not formed, offering the possibility to probe the potential of UAS-driven wild type or mutant Abd-B protein expression to restore genital disc development. To quantify the extent of this rescue we counted the number of vaginal teeth (Fig 7F). Expression of the wild type Abd-B protein allowed for a very mild rescue, with no teeth in most cases and reaching 7 teeth just in a few individuals (Fig 7E and 7F), while the wild type fly harbors around 28 vaginal teeth (Fig 7A). This poor rescue may be due to this Gal4 line not reproducing precisely in time and space the expression of the wild type *Abd-B*. By comparison, the Abd-B$^W$ has a slightly better rescuing ability (Fig 7E and 7F). We next investigated the requirement of Abd-B protein motifs already probed in the context of Abd-B A7 suppressive function. As stated above, the wild type protein has a very poor genital disc rescuing ability, as measured by vaginal teeth number, only allowing to identify improved activity, which was the major effect seen in the rescue of A7 suppression (Fig 6). Results identified the TG (within the HX like motif region), KK (within the HD N-terminal arm) and QR (C-Terminal to the HD) mutations as resulting in a major increase of Abd-B genital disc rescuing ability (Fig 7E and 7F).

We also probed intrinsic protein requirements for Exd-independent functions. We focused on the wing imaginal disc where Exd is cytoplasmic and therefore inactive [67], investigating in gain of function experiments the expression of *wingless* (*wg*). *wg* is expressed encircling the wing pouch and in the dorso-ventral boundary in the wing disc but not in the posterior compartment of the haltere disc, due to *Ultrabithorax* repression [68,69]. Abd-B protein expression (wild type or mutated version) was driven in the posterior compartment by the *hh*-Gal4 line [70], where its repressive effect on *wg* expression (*wg-GFP* reporter line) was monitored. Results identified a very strong impact of the KK mutation, with the Abd-B$^{KK}$ protein displaying increased *wg*-GFP repressive activity. The TG, YPWM and CEN variants also demonstrate increased *wg*-GFP repressive activity (S5 Fig). Increased activity of these Abd-B mutant forms suggests that these residues mediate interactions with an unknown factor restricting Abd-B repressive potential in the wing pouch.

The study of intrinsic protein requirements for Abd-B function in the three situations investigated (male A7 suppression, female genital disc (vaginal teeth) development and wing-pouch *wg* repression) shows a clear directionality of observed effects with mutations in all cases driving increased Abd-B activity. This indicates a prominent role of protein activity buffering, limiting Abd-B activity. We do not know whether these sequences directly mediate contacts towards Exd (and/or Hth). However, the fact that Exd and Hth exert a similar buffering activity suggests that they may do so. If so, and in the case of the HX-like motif, the mode of interaction is likely different from the canonical HX motif, since the central core W residue in Abd-B does not contribute to Abd-B A7 suppressive function while the surrounding residues do. The data also highlight a context specific use of the protein sequences under study: some mutations (TG and QR) impact on the three functions, others (YPWM, KK, CEN) impact two of the three functions and the EWTG mutations specifically impact on male A7 suppressive function.

## The central W residue of the Abd-B HX-like motif is required for Abd-B function in the central nervous system

The central W residue of the HX-like motif is evolutionary conserved, arguing for functional importance, which contrast with the lack of clear functional requirement. We thus aimed at investigating other developmental functions of Abd-B that could reveal a function for this residue.

Given the contextual use of protein motif described above, we reasoned that the W residue conservation might result from an Exd-dependent function in another tissue than the male

abdomen and female genitalia, and examined Abd-B function in the embryonic nervous system. A subset of 30 neuroblasts (NBs) found in each hemi-segment generates the embryonic ventral nerve cord, where Hox genes control segment-specific differences. This is well illustrated by the NB6-4 progenitor, which generates neuronal and glial cells in the thoracic segments, while producing only glial cells in the abdomen. The lack of neuronal cells derived from NB6-4 results from repression by the Hox genes *abd-A* (anterior abdomen) and *Abd-B* (posterior abdomen) [71]. The abdominal specific NB6-4 lineage was shown to depend also upon Exd and Hth [72]. In the case of *abd-A*, it was further shown that an Abd-A/Exd/Hth complex binds to a *cycE* (*cyclin E*) cis regulatory element mediating transcriptional repression of this gene in NB6-4 [71,72]. Forcing *abd-A* expression in the thoracic segments (where Exd and Hth are already present) was shown to transform the thoracic NB6-4 lineage (neuron + glia) towards an abdominal lineage (only glia). We have tested if the ectopic expression of *Abd-B* does the same. Wild type or W mutated Abd-B was expressed using the *scabrous* (*sca*)-Gal-4 driver, active from early stages in all neuroblasts, including in thoracic NB6-4. We used Eagle as a marker to identify the NB6-4 lineage (neuron + glia) [73], Repo as a general marker of glial cells [74,75] and the 1.9 *cycE* enhancer shown to recapitulate the Hox-dependent *cycE* expression in NB6-4 [71]. Results show that *Abd-B* promotes a thoracic to abdominal transformation of NB6-4, evidenced by the lack of NB6-4/CycE positive cells in the thorax (Fig 8A and 8B). By contrast, the expression of Abd-B$^W$ does not promote such a transformation (Fig 8A and 8B), indicating the functional requirement of the W conserved residue of the HX-like motif for Abd-B function in controlling segment specificity in the embryonic ventral nerve cord.

We next investigated if, as shown for Abd-A [72], Abd-B forms a complex with Exd and Hth on DNA elements that mediated Hox repression. Electrophoretic mobility shift assays shows that Abd-B forms a trimeric complex with Exd/Hth on the *cycE* cis-regulatory element (Fig 8C). Mutation of the W residue affects this trimeric complex with more than half (56%) reduction, while mutation of the QR sequence leads to a 25% reduction (Fig 8C). Further supporting synergistic DNA binding, we also found that Abd-B and Exd/Hth form a complex on the *Dll*$^{con}$, a consensus sequence previously shown to bind most Hox proteins [76] (Fig 8C). However, on this specific DNA sequence, the W mutations increases Abd-B monomeric binding, while not affecting the binding of the trimer (Fig 8C). Taken together with the antagonistic effect Exd/Hth exert on Abd-B binding on DIIR, the sequence mediating repression of the limb promoting gene *Distal-less* (*Dll*; 51, 76), this indicates that both the nature of Abd-B interaction with Exd/Hth and its dependency upon the HX central core W residue depends on the identity of the DNA target sequence. We concluded that in the embryonic ventral nerve cord, Abd-B acts in a manner similar to anterior and central Hox proteins, through synergistic DNA binding with Exd and Hth requiring the HX like motif and, specifically, the W residue.

We also probed the contribution of the Abd-B QR sequences, located in a position shown to mediate direct contact with Exd in other central and anterior Hox proteins [24,44]. In a manner similar to the W mutation, the QR mutation alleviates Abd-B thoracic to abdominal NB6-4 lineage transformation and impacts the potential of Abd-B to form a trimeric complex specifically on the *cycE* enhancer sequences. This suggests that the mode of cooperative binding used by Abd-B, with sequences both upstream and downstream of the HD, is similar to that described for central and anterior Hox proteins.

The posterior Hox proteins Ubx, Abd-A, and Abd-B were also shown to repress the expression of *eyes absent (eya)*, which is specifically expressed in thoracic neurons. This repression is Exd and Hth dependent [77]. Forced thoracic expression of Abd-B, but not Abd-B$^W$, results in the lack of thoracic specific *eya* neurons (S6 Fig). Similar to the effect on the NB6-4 lineage, Abd-B thus promotes an abdominal transformation that is dependent upon the HX like W

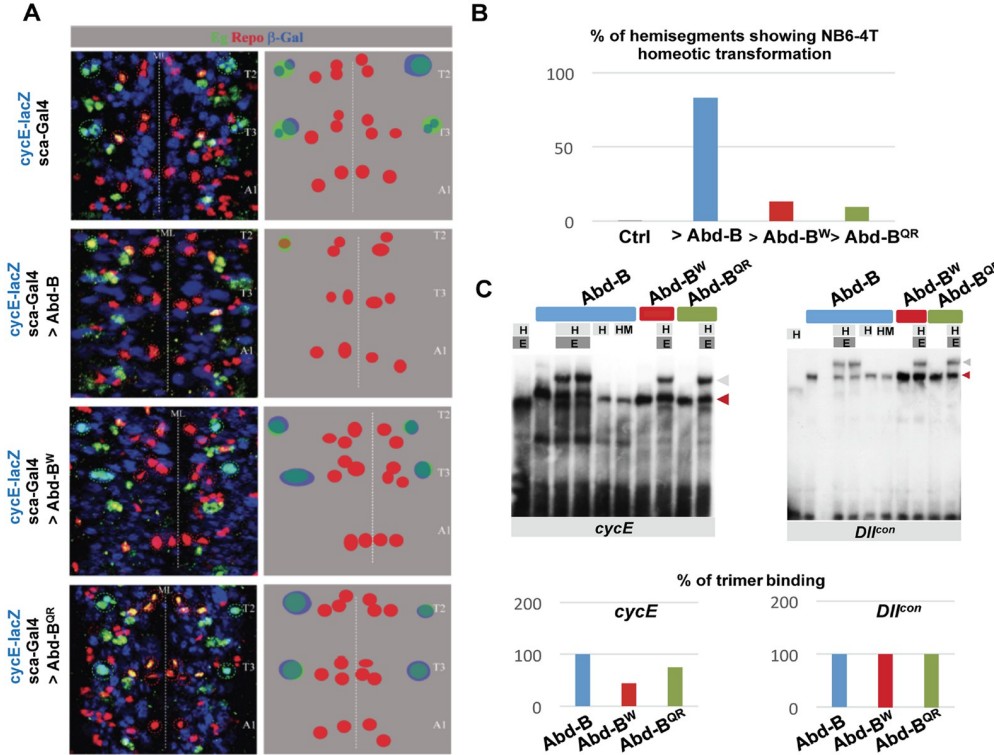

**Fig 8. Functional synergy of Abd-B and Exd/Hth in the embryonic CNS. (A)** Impact of thoracic expression of Abd-B, Abd-B$^W$ or Abd-B$^{QR}$ on the NB6-4 CNS lineage. Thoracic expression was achieved using the *scabrous* (*sca*) Gal-4 driver, active from early stages in all neuroblasts, including thoracic NB6-4. Eagle (Eg; in green) was used as a marker to identify the NB6-4 lineage (neuron + glia) and Repo (in red) as a general marker of glial cells. The 1.9 *cycE* enhancer, shown to recapitulate the Hox-dependent *cycE* expression in NB6-4, was monitored to assess the impact of Abd-B W and QR mutations on *cycE* expression and NB6-4 abdominal lineage specification. Abd-B promotes a thoracic to abdominal transformation of NB6-4 evidenced by the lack of NB6-4/CycE positive cells in the thorax, a transformation that requires the integrity of the W and QR residues. T2 and T3 indicate second and third thoracic segments, and A1 the first abdominal one. ML, midline. **(B)** Quantification of the Abd-B, Abd-B$^W$ and Abd-B$^{QR}$ abdominal homeotic transformation, assessed by the percentage of hemisegments showing NB6-4 homeotic transformation. **(C)** EMSA experiments assessing the impact of the W and QR mutation on the formation of an Abd-B/Exd/Hth/DNA complex, using the *cycE* and *Dll$^{con}$* targets. While the W and QR mutations do not affect Abd-B/Exd/Hth/DNA complex formation on the *Dll$^{con}$* sequence, they do so on the *cycE* DNA target. E and H refer to Exd and Hth, respectively, and HM to the N-ter domain of Hth that mediate contact with Exd (10).

central residue. The lack of identified cis-regulatory region mediating the Abd-B repression does not allow addressing if this repression also involves the assembling of an Abd-B/Exd/Hth complex on the *eya* cis-regulatory region.

## Discussion

### Gene cross-regulatory interactions define different types of Abd-B-Exd/Hth functional antagonisms

Our study of *Abd-B* and *exd*/*hth* gene regulation highlights a tight reciprocal control: an increase in *Abd-B* down-regulates *exd*/*hth* expression, a reduction of *hth* diminishes *Abd-B*, whereas up-regulation of *exd*/*hth* reduces Abd-B activity (Fig 9). This results in the co-existence of Abd-B and Exd/Hth in the male A7, allowing for functional interactions at the protein level. In the male A7, excess of Exd/Hth antagonizes Abd-B A7 suppressive activity. This

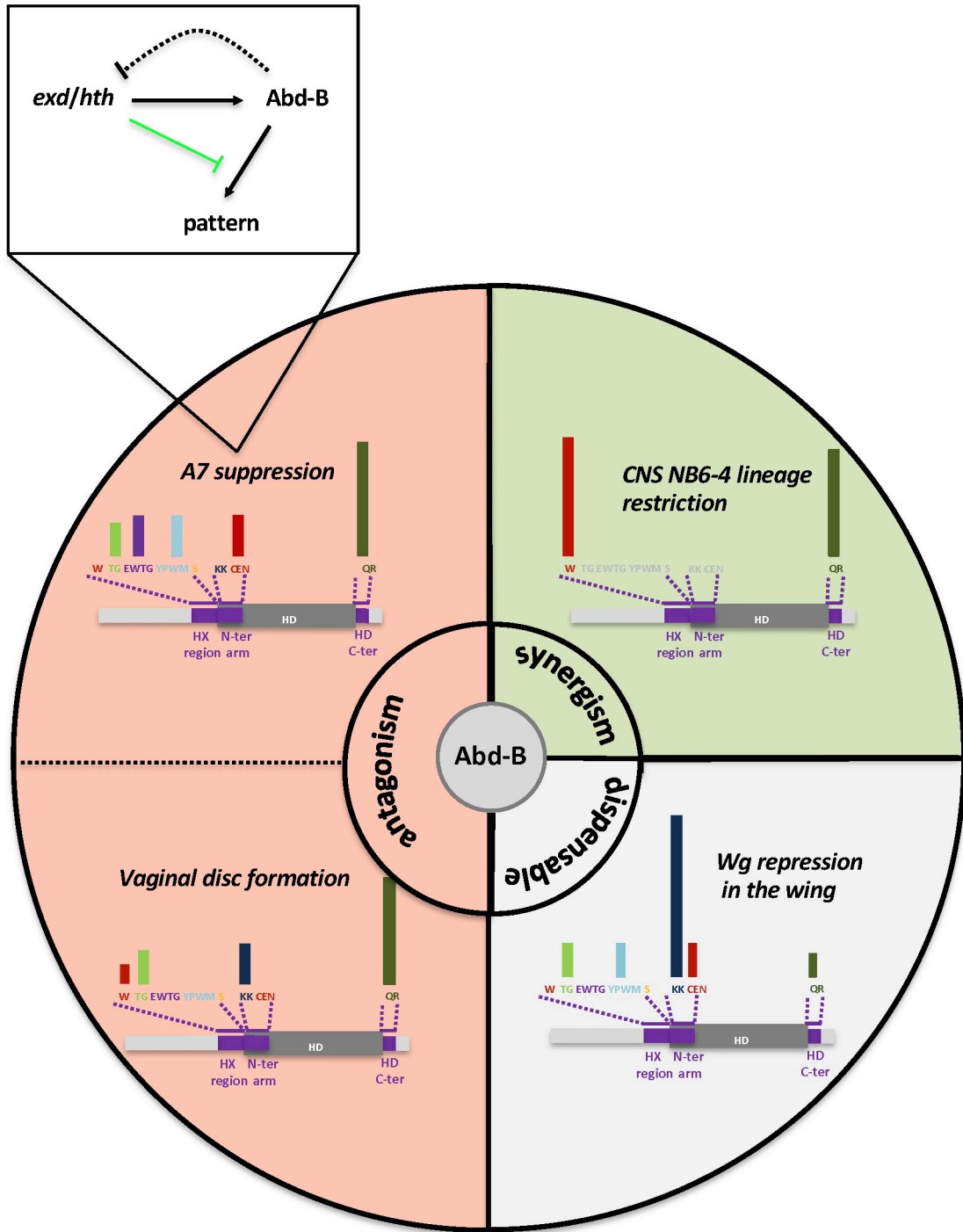

**Fig 9. Summary of Abd-B partnership with Exd/Hth and intrinsic protein requirements.** Within the circle, functional antagonism is highlighted in red, synergism in green, and dispensability in grey. Protein intrinsic requirement are summarized for each of the four developmental contexts illustrated. Bars above the scheme of Abd-B protein are a qualitative representation of protein sequence requirement. Non-assessed protein sequences are in grey. A functional relationship between Exd/Hth and Abd-B for A7 suppression is indicated.

precise functional interplay is dose sensitive, underscoring the importance of protein dosage, which was also proposed for Ubx protein function, with developmental and evolutionary implications [78–80].

Functional antagonism between Abd-B and Exd/Hth was previously described in the embryonic A8 segments, where *Abd-B* promotes the development of the posterior spiracle [50] and represses the limb-promoting gene *Dll* [51]. In both instances, *Abd-B* downregulates the expression of *exd*/*hth*, resulting in the lack of Abd-B and Exd/Hth co-existence in the embryonic A8 segment. When Exd and Hth are maintained in this segment, they antagonize the activity of Abd-B, impairing posterior spiracle development and *Dll* repression [50, 51]. The study of protein binding to a *Dll* regulatory element that recapitulates *Dll* repression by Abd-B, and to a regulatory element of the *empty spiracle* gene, an Abd-B direct downstream target essential for posterior spiracle development, further indicated that Exd/Hth compete out Abd-B binding to these regulatory elements [50,51]. This was further proposed to result from Abd-B and Exd/Hth direct protein interactions, which we also observed *in vivo* in the pupal A7 segment using BiFC and in SF9 co-precipitation assays. The lack of identified Abd-B direct target sequences mediating Abd-B function in A7 suppression, however, prevents addressing if Exd/Hth counteract Abd-B binding to downstream targets key to A7 elimination. Functional antagonism may also apply in the female genitalia, where Abd-B promotes vaginal disc formation, while Exd/Hth suppresses it [66].

## Synergism versus antagonism: Mechanisms and Hox class specificity

Molecular mechanisms beyond the Abd-B and Exd/Hth interplay may, in principle, follow two scenarios: Abd-B and Exd/Hth may physically converge on cis-regulatory sequences of common target genes, or Abd-B and Exd/Hth may act on different target genes that functionally converge. Discriminating between these two possibilities will require the identification of Abd-B and Exd/Hth direct target genes and the dissection of their molecular control, this being a pre-requisite for a comprehensive understanding of Abd-B and Exd/Hth interplay, either synergistic or antagonistic. Our results do not exclude, nevertheless, that Abd-B may regulate some genes as a monomer or that Exd/Hth may have an Abd-B-independent function.

The function of Abd-B in the embryonic CNS provides support for Abd-B-Exd/Hth functional synergism. In a manner very similar to Abd-A, Abd-B forms a trimeric complex on a *cycE* regulatory element, indicating that Abd-B and Exd/Hth cooperative DNA binding mediates functional synergism. Such a functional synergism may also apply to the regulation of the *eya* gene in the embryo, where Abd-B and Exd/Hth are required for repression or gene activation [71,72].

Taken together with the above-discussed antagonistic function, Abd-B displays both synergistic and antagonistic relationships towards Exd/Hth. The mechanisms channeling Abd-B-Exd/Hth towards synergistic versus antagonistic actions are at least dictated by the identity of the DNA target sequence. This is illustrated by in vitro band shift assays showing that Exd-Hth counteract Abd-B binding to a short *Dll* regulatory element [51], while it cooperatively binds to a slight modification (4 bp) of the same regulatory element (this study). Differences in a cell's protein content likely may also contribute, which may direct Abd-B and Exd-Hth to different functional outputs in different cellular context, tissues, or developmental windows. Although the cooperative or antagonistic effect of Abd-B and Exd/Hth depends on the target, the anteriorwards transformation of A6 and A7 abdominal segments when *exd* or *hth* are up-regulated, including changes in phenotypic traits like formation of trichomes or, most likely, pigmentation, suggests common antagonistic effects on several genes.

The potentiality of the posterior Hox protein Abd-B to act both synergistically or antagonistically with Exd/Hth is clearly distinct from the widely supported synergistic functional relationship with anterior and central Hox proteins [17]. We note, however, that Exd-Hth and the central Hox protein Ubx act antagonistically in *Drosophila* muscle fiber identity specification, with Exd-Hth promoting and the central Hox protein impairing fibrillar (flight muscle) specification [81]. It was shown that binding of an Exd-Hth complex on *Actin88F (Act88F)*, a key player in fibrillar muscle fate specification, promotes *Act88F* expression and fibrillar muscle identity. In the presence of the central Hox protein Ubx, a trimeric Ubx-Exd-Hth complex assembles on *Act88F*, leading to *Act88F* repression and the establishment of tubular muscle fate as a default state [81]. Although this is a clear case of functional antagonism, it still relies on DNA binding cooperativity, indicating that the output of cooperative DNA binding may be either functional synergism or antagonism.

## The protein landscape of Abd-B functions

We investigated the impact of Abd-B mutations spanning the HX region preceding the HD, those in the HD N-terminal arm and in the regions immediately downstream the HD, defining a complex pattern of protein domain requirements (Fig 9).

The HX central W residue appears dispensable for Abd-B and Exd/Hth antagonistic activities in A7 segment suppression. Contrasting with the lack of effects of the W mutation, mutations of residues surrounding the W impact A7 suppressive activity, suggesting a mode of Exd/Hth interaction, within the HX region, that may differ from that described for anterior and central Hox proteins, where the central W residue is essential. Contrary to what occurs in the male A7 and female genitalia, the HX central W residue is essential for synergistic repression of the NB6-4 lineage and *cyc-E* expression. This indicates that in synergizing with Exd/Hth, Abd-B seems to use similar molecular modalities than anterior and central Hox proteins, with a key contribution of the HX core W residue, while it uses different modalities, relying on W surrounding residues, for functional antagonism.

The Abd-B HX surrounding sequences and the HD N-terminal arm appear required for Abd-B functions whenever tested. This includes Abd-B functions that rely both on Abd-B and Exd/Hth (male A7 suppression and female genitalia formation), or functions where Abd-B acts in the absence of Exd/Hth (*wg* repression in the wing imaginal disc). Yet, the precise contribution of different residues within each of these regions appears dedicated to different Abd-B functions: for example the CEN residues do not contribute to female genitalia formation but do so to male A7 suppression and, reversely, the KK residues contribute to female genitalia formation but not to male A7 suppression.

Sequences immediately C-terminal to the HD appear essential for all Abd-B functions that are Exd/Hth dependent, irrespective of whether Abd-B and Exd/Hth display functional antagonism or synergism. This is illustrated by a strong contribution to male A7 suppression, female genitalia formation and CNS NB6-4 lineage restriction, while only marginally required for *wg* repression in the wing disc that is Exd/Hth independent.

Overall, the data identify distinct effects of Abd-B protein mutations for different Abd-B functions, highlighting that a mutation often affects only a subset of Abd-B functions. This was also observed for the *Drosophila* Abd-A protein, where variable requirements of protein domain for different Abd-A functions were reported [82]. Additional studies have also shown the effect of mutations in some protein motifs are pleiotropic, with some mutations affecting one character and not others [82–87]. Differential pleiotropy thus seems to apply to Hox proteins in general and highlights that molecular modalities of Hox protein activity are context specific. The precise understanding of Hox protein mode of action, including how they

interface with Pbc/Meis cofactors thus requires detailed mechanistic analysis of a larger number of Hox protein functions.

## Material and methods

### Genetics

Vallecas strain was used as a wildtype. *Abd-B^{M5}* is a null mutation for the Abd-Bm isoform and *Abd-B^{M1}* is a strong loss-of-function *Abd-B* allele for all Abd-B functions [46,48]; *hth^{P2}* is a strong loss-of-function *hth* allele [32,88,89]; Dp(3;3)P5 is a tandem duplication for the three genes of the Bithorax Complex [59]; Wg-GFP is a Wg protein trap [61].

The Gal4/UAS system [90] was used to express ectopically genes or to inactivate them with UAS RNAi constructs. The Gal4/Gal80^{ts} system [91] was used to control the time of activation or inactivation of different genes. In experiments in which larvae or pupae were shifted to 29˚C care was taken to take into account the different developmental time at this temperature compared to that at 25˚C. We estimated developmental time at 29˚C as about 25% faster than at 25˚C so that the hours after puparium formation (APF) given are an approximation to the developmental time at 25˚C.

**Gal4 lines.** *Abd-B^{MD761}* is a Gal4 insertion in the *iab-7* regulatory region of *Abd-B* that drives expression in the A7 segment and is also mutant for the *iab-7* function (named as *MD761*-Gal4 in 53). *Abd-B^{LDN}* is an insertion in the *Abd-B* gene that reproduces its expression in the female genital primordium (A8) and is also mutant for the *Abd-Bm* function (65; *Abd-B-Gal4^{LDN}* in that work). *hh*-Gal4 drives Gal4 expression in the posterior part of the wing disc [70]. *sca*-Gal4 (BS 5479) directs expression in all neuroblasts [92] and *elav*-Gal4 (elavC155--Gal4, BS 458) in neurons [93].

**UAS lines.** UAS-Abd-BM 1.1 [60], and UAS-Abd-BM (inserted in the ZH35 landing site; 51), UAS-*GFP* [94], UAS-*hth* RNAi (Vienna *Drosophila* RNAi Center, VDRC, 12763, second chromosome and 12764, third chromosome), UAS-*exd* RNAi (VDRC, lines 7802 second chromosome and 7803, third chromosome), UAS-*Abd-B* RNAi (VDRC, line 12024), UAS-*cherry* (BS 35787), UAS-*hth* [32], UAS-*exd* [50,51] UAS-Abd-B^{VC} and UAS-^{VN}Exd [39]. The different UAS-Abd-B lines were described in ref. 51. Unless otherwise indicated, the UAS-Abd-B line inserted in ZH35 was the one used in the different experiments.

In several experiments we used the Ga4/UAS/Gal80^{ts} system [91] to express different genes during the third larval instar or white pupa stages onwards by shifting the larvae or pupae from 18˚C to 29˚C. In this way we prevented the lethality, poor viability or abnormal development of different mutant combinations. In the experiments in which we expressed with the *pannier*-Gal4 line different UAS constructs, expressing or inactivating *exd*, *hth* or *Abd-B* (including the BiFC experiments), we shifted the larvae from 18˚C to 29˚C at the third larval instar and observed pupae normally at about 22-26h APF at 29˚C, which we calculated roughly corresponds with about 28-32h a 25˚C. In the analysis of rescue of the A7 mutant background (*Abd-B^{MD761}*/*Abd-B^{M1}*) with the different UAS-Abd-B variants we also used the Gal4/Gal80^{ts} system and shifted white pupae from 18˚C to 29˚C. The experiments in which we drove expression of the Abd-B variants under the control of the *hh*-gal4 driver in the wing disc, also with *tub*-Gal80^{ts}, the shift from 18˚C to 29˚C was done in second instar or early third instar larvae and fixation of the discs in late third instar larvae. Controls in the different experiments underwent the same temperature treatments.

### Clonal analysis

We used the FLP/FRT system [95,96] to make clones mutant for *hth^{P2}* or *Abd-B^{M5}*. Clones were induced with the hs-flp construct and an hour heat-shock at 37˚C during the third larval

period (although histoblasts do not divide until pupa) and marked by the absence of GFP expression. Flp-out clones [97], marked by the presence of GFP expression, were induced during the third larval period by setting the vials at 37˚C for 10–15 minutes to induce the expression of *hth*, *hthRNAi* or *Abd-B*. The chromosomes used to induce clones were FRT82B *ubi*-GFP, FRT82B *hth*$^{P2}$ (gift from N. Azpiazu, CBMSO, Madrid), and FRT82B *Abd-B*$^{M5}$ [46] For flip-out clones, *act>y+>Gal4 UAS GFP* [94] was used. The genotypes of the larvae where clones were induced were:

*Abd-B* mutant clones: *y hs-flp122/y w o Y; FRT82B Abd-B*$^{M5}$*/FRT82B ubi-GFP*

*hth* mutant clones: *y hs-flp122/y w o Y; FRT82B hth*$^{P2}$ */FRT82B ubi-GFP*

*hth* flip-out clones: *y hs-flp122/w o Y; act> y+>Gal4 UAS- GFP/UAS-hth RNAi*, or UAS-*hth*

*Abd-B* flip-out clones: *y hs-flp122/w o Y; act> y+>Gal4 UAS- GFP/UAS-Abd-B*

## Analysis of adult cuticles

Flies were dissected and macerated in a 10% KOH solution at 100˚C to remove the internal structures. The cuticles were mounted in glycerol, adding a small amount of Tween-20.

## Immunohistochemical analysis

Fixations, staining, dissection and mounting of the pupae (male) were done following published protocols [53,98]. Pupae were visualized in a Zeiss LSM510 and Nikon A1R confocal microscopes. The primary antibodies used were: mouse anti-Abd-B (Developmental Studies Hybridoma Bank, (DSHB), Universidad de Iowa), used at 1:10–1:50, rat anti-Exd (gift of N. Azpiazu, CBMSO, Madrid), (used at 1:100), guinea pig anti-Hth (used at 1:100) (gift of N. Azpiazu, CBMSO, Madrid), rabbit anti-GFP (Invitrogen), (1:300). Secondary antibodies were Alexa 488, Alexa 555, and Alexa 647 of the corresponding species (Invitrogen), 1:200. To stain nuclei we used To-Pro 3 (Invitrogen), 1:1000.

Embryo fixing and staining for NB6-4 lineage analysis was as described in ref. 72. Primary antibodies used were mouse or rabbit anti-ß-gal (1/1000, Cappel), anti-Repo (mouse, 1/10 DSHB) and anti-Eg (rabbit, 1/500, gift of L. S. Shashidhara). The 1.9 kb CycE-lacZ reporter construct strain is defined in [99]. *sca*-Gal4, CycE lacZ females were crossed to males of UAS lines. Crosses were maintained all times at 30˚C at regular fly culture conditions.

## Bimolecular fluorescence complementation

We have used this method to study interactions between Abd-B and Exd in the dorsal pupal epidermis with the GAL4/UAS method using the *pannier* (*pnr*)-Gal4 line and the Gal4/Gal80$^{ts}$ system, following the methods described before [64]. The lines used were UAS-Abd-B$^{VC}$, UAS-Abd-B$^{(W>A)VC}$ and $^{VN}$UAS-Exd [39,100]. Larvae were shifted from 18˚C to 29˚C as third instar larvae. Quantification of identical areas in the *pnr*$^+$ and *pnr*$^-$ domains were done as described below. A minimum of 10 pupae was used to obtain each result.

## Quantification of expression levels

To compare levels of expression of Abd-B of the different Abd-B variants in the experiment of rescue of the A7 segment mutant phenotype, we transferred white pupae of the genetic combination UAS-Abd-B (X) *Abd-B*$^{MD761}$ UAS-GFP *tub*-Gal80$^{ts}$ *Abd-B*$^{M1}$ from 18˚C to 29˚C, let the pupae develop for 24-28h, fixed and stained them with an anti-Abd-B antibody. We

selected an area of 10 nuclei (horizontally) per 4 nuclei (vertically) in the posterior part of the A7a and A6a segments, quantified Abd-B levels in both segments with Image J software and obtained the A7/A6 ratio. At least ten pupae were quantified to obtain mean and standard deviation. In this and the rest of quantifications, shifting of larvae or pupae, dissection, fixing and antibody staining were done with the same experimental conditions. Acquisition of the confocal images was also done under identical conditions.

To compare repression of wg-GFP and levels of expression of Abd-B of the different variants when expressed ectopically in the wing disc, we have used the *hh*-Gal4 line [70], which drives expression in the posterior part of the wing disc. We stained the discs with anti-Abd-B and quantified the GFP and Abd-B expression levels with Image J software and the Measure tool. For wg-GFP we selected a small rectangular area encompassing the wg-GFP band of expression at the dorso-ventral boundary in the posterior compartment, adjacent to the A/P compartment boundary, and the same area in the corresponding region of the anterior compartment, also adjacent to the A/P boundary. We measured GFP levels with the Measure tool of image J and divided anterior by posterior levels. A minimum of 10 larvae was used to obtain each result.

In different genetic combinations the levels of Abd-B, Exd or Hth were quantified in pupae also with Image J software. To compare expression levels in the pupal abdomen we used the *pannier* (*pnr*)-Gal4 line, which is driving expression in the central part of the abdomen. The expression domain is labeled with GFP and we have measured, in each case, the expression of Abd-B, Exd or Hth proteins in identical areas in the $pnr^+$ and $pnr^-$ domains (adjacent areas). The measures were taken with Image J software and the relative expression of the $pnr^+/pnr^-$ domains was calculated.

## Phenotype normalization

For each variant, the ratio Abd-B variant/Abd-B was determined in the male A7 segment and wing pouch. Given that measurement of protein levels in female genitalia are not possible in our mutant conditions, we considered that, because the same UAS-lines were used, the relative expression in the male A7 and female genitalia would be similar (even if absolute levels may differ). The ratio obtained from measurement in the male A7 for each line was thus also taken for correcting vaginal teeth counts in the female genitalia. Correction was operated by dividing values per Abd-B variant/Abd-B mean ratio for wg-GFP and vaginal teeth counts, and multiplying values per Abd-B variant/Abd-B mean ratio for A7 bristle counts. Statistical analysis for protein variant activities was performed using the Wilcoxon test (****$p<0.0001$; $0.0001<$***$p<0.001$; $0.001<$**$p<0.01$; $0.01<$*$p<0.05$).

## Co-immunoprecipitation

The vectors used for the co-immunoprecipitation experiments contain cDNAs encoding the Abd-B (wildtype and W>A variant), Exd, and Hth proteins fused with hemagglutinin (HA; HA-Abd-B, HA-Abd-B$^W$) Histidine (His; Exd-His) and Flag (Hth-Flag), respectively. They have been subcloned by conventional procedures in the pFastBac vector (all subclones have been sequenced to verify that they contained the appropriate mutations), and the protocol recommended by Invitrogen (Bac-to-Bac Expression System) has been followed to elaborate baculoviral vectors and induce protein expression. The cells have been collected by centrifugation, washed with cold PBS and resuspended in the lysis solution (140 mM KCl, 5% glycerol, 2.5% MgCl2, 25mM Hepes pH 7.8, Protease inhibitor, 10mM imidazole). Protein expression levels have been quantified by Western Blot, taking these measures into account to mix Exd-Hth with the different Abd-B mutant variants, adding lysis solution where necessary to obtain

the same final volume. The immunoprecipitation experiments have been carried out in 2 ml vials, with 200 microliters of anti-Flag resin (Sigma-Aldrich) and 580 microliters of extract. We followed the protocol recommended by the commercial company to carry out the co-immunoprecipitation experiments. 10 μl of 580 μl of input and 10 μl of 200 μl of elution were loaded on gels. The results were visualized by Western Blot, using mouse anti-HA (Sigma-Aldrich), mouse anti-His (Abcam) and rabbit anti-Flag (Sigma-Aldrich) as primary antibodies. Secondary antibodies have been anti-mouse or anti-rabbit coupled to specific IgG light chain HRPs (Jackson). Exposure was made using ECL solution and photographic film. Bands were quantified using the Fiji gel analysis tool. A correction factor (0,34) taking into account the difference in the total volume of Input and Elution fractions was applied when calculating the Elu/Input ratios. The percentage of tryptophan-induced loss of binding to Hth was calculated as follows: for Abd-B: ([bound Abd-B–bound Abd-B$^W$]/bound Abd-B) x 100; for Exd: ([bound Exd when Abd-B is used–bound Exd when Abd-B$^W$ is used] / bound Exd when Abd-B is used) x 100. Input fractions were pre-run to allow for equally loaded elution fractions.

### EMSA experiments

Abd-B proteins for EMSA were produced using the TNT coupled in vitro transcription/translation system (Promega). Protein production was estimated by labeling the proteins with $^{35}$S-methionine; the proteins were found to be produced at similar amounts. EMSAs were performed in 20 μl as described [101] using radiolabelled DllR$^{con}$ [76] or cycE probes [72].

### Image acquisition

All confocal images were obtained using Zeiss LSM510 and Nikon A1R vertical confocal microscopes. Image treatment and analysis was performed using Image J, Adobe Photoshop and Powerpoint softwares.

### Measurements and statistical analysis

To compare between two groups, as was the case with comparing wildtype Abd-B and the Abd-B$^W$ variant (in bristle number, expression levels, BiFC signal), for Abd-B, Hth and Exd levels in different mutant combinations, and for bristle number in comparing flies with different doses of Abd-B and Exd, the two-tailed non-parametric Student's *t*-test test was used. To compare between more than two groups (wg-GFP repression in larvae expressing combinations of Abd-B and Exd proteins or controls) a non-parametric, one-way ANOVA Dunnett's test was used. Symbols to indicate significance were: ****$p < 0.0001$; $0.0001 < $***$p < 0.001$; $0.001 < $**$p < 0.01$; $0.01 < $*$p < 0.05$. In all cases the Graph Pad Prism software was used. Raw data for all quantified experiments are displayed in S1 Table.

### Supporting information

**S1 Fig. Expression of *exd* and *hth*.** Expression of Exd (**A, A'**) and Hth (**B, B'**) in *Abd-B$^{MD761}$* UAS-GFP male pupa of about 28-30h APF, showing similar levels of expression of Exd and Hth in the A7 and A6 segments. *Abd-B$^{MD761}$* is a Gal4 line driving expression in the A7; see Methods and main text.
(TIF)

**S2 Fig. *Abd-B* mutant clone up-regulating *exd* and *hth*.** Big clone mutant for *Abd-B$^{M5}$* induced in the male A7 segment and marked by the absence of GFP, showing slightly increased levels of Exd (in red) and Hth (in blue) with respect to most adjacent cells.
(TIF)

**S3 Fig. *Abd-B*-expressing clones in the A6 and A7. (A)** Clone expressing *Abd-B* and marked with GFP in the A6. The expression of Exd (red) or Hth (grey) is not modified with respect to surrounding cells. (**B**) Clones expressing *Abd-B* and marked with GFP in the A7. The expression of Exd (red) or Hth (grey) is slightly reduced (except for one nucleus) with respect to surrounding cells in two clones (green arrows) but only in two nuclei in another clone (red arrow).
(TIF)

**S4 Fig. Combined inactivation or expression of *exd* and *hth*. (A)** The reduction of *exd* expression in the male A7 transforms it partially to A6, and this transformation is not increased by the simultaneous reduction of *exd* and *hth* expression in this segment **(B)**. **(C, D)** The increase of either *exd* alone (C) or *exd* and *hth* (D) in the male A7 results in the development of a small segment of similar size in both genotypes.
(TIF)

**S5 Fig. Repression of *wg* by Abd-B variants in the wing disc (A-J)** The different panels show wg-GFP expression in wing discs of the *hh*-Gal4 *tub*-Gal80$^{ts}$/+ genotype and of *hh*-Gal4 *tub*-Gal80$^{ts}$ driving expression of the different Abd-B proteins. In all cases the larvae were transferred in second or early third instar larvae from 18° to 29°C. In the wing pouch the wg-GFP line directs GFP expression, as *wg*, in two rings around the pouch and in a dorso-ventral band. **(K)** Quantification of Abd-B expression driven by *hh*-Gal4 in the posterior wing pouch. The wild type and Abd-B variant proteins do not show significant differences in expression levels (n = 10–15 for the wildtype Abd-B and all the variants). **(L)** Normalized wg-GFP expression following expression of Abd-B and Abd-B variants in the posterior wing pouch. Measurements were taken at the D/V boundary, both in the A (no Abd-B variant expression) and P (Abd-B variant expression) compartments (see methods); (n = 10–13 for the wildtype Abd-B and all the variants). The A/P ratio of wg-GFP expression is plotted and was normalized relative to the level of Abd-B and Abd-B protein variant expression (counts / ratio Abd-B variants/ Abd-B). Significant differences identify gain in Abd-B repressive activity for Abd-B$^{TG}$, Abd-B$^{YPWM}$, Abd-B$^{KK}$, Abd-B$^{CEN}$ and Abd-B$^{QR}$.
(TIFF)

**S6 Fig. Repression of *eya* by wildtype Abd-B and Abd-B$^{W}$. (A)** Wildtype Eya expression. **B**). Forced *elav*-driven expression of *Abd-B* in the thorax results in the lack of thoracic specific Eya neurons. **(C)** Mutation of the W residue alleviates the repression in these neurons.
(TIF)

**S1 Table. The table displays for each of the quantitative figure panels the underlying data.**
(XLSX)

**S1 Text. List of the genotypes in each figure.**
(DOCX)

## Acknowledgments

Institutional support from Fundación Ramón Areces to the CBMSO is acknowledged. We thank N. Azpiazu, G. Morata and L. S. Shashidhara for stocks and antibodies, Nuria Prieto for aiding in some experiments, the Confocal Microscopy Service at the Centro de Biología Molecular Severo Ochoa for help with the experiments, and Flybase, the Bloomington Stock

Center, the Vienna *Drosophila* Resource Center and the Developmental Studies Hybridoma Bank (University of Iowa) for information, stocks and antibodies.

## Author Contributions

**Conceptualization:** Jesús R. Curt, Ramakrishnan Kannan, Samir Merabet, Andrew J. Saurin, Yacine Graba, Ernesto Sánchez- Herrero.

**Data curation:** Jesús R. Curt, David Foronda, Bruno Hudry, Ramakrishnan Kannan, Samir Merabet, Andrew J. Saurin, Yacine Graba, Ernesto Sánchez- Herrero.

**Formal analysis:** Jesús R. Curt, David Foronda, Bruno Hudry, Ramakrishnan Kannan, Samir Merabet, Andrew J. Saurin, Yacine Graba, Ernesto Sánchez- Herrero.

**Funding acquisition:** Ramakrishnan Kannan, Yacine Graba, Ernesto Sánchez- Herrero.

**Investigation:** Jesús R. Curt, Paloma Martín, David Foronda, Bruno Hudry, Ramakrishnan Kannan, Srividya Shetty, Ernesto Sánchez- Herrero.

**Supervision:** Ramakrishnan Kannan, Yacine Graba, Ernesto Sánchez- Herrero.

**Validation:** Samir Merabet, Andrew J. Saurin, Yacine Graba, Ernesto Sánchez- Herrero.

**Writing – original draft:** Yacine Graba.

**Writing – review & editing:** Samir Merabet, Andrew J. Saurin, Yacine Graba, Ernesto Sánchez- Herrero.

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
