## [Decision Letter · Decision Letter 0]

24 Aug 2024

Dear Dr Sánchez-Herrero,

Thank you very much for submitting your Research Article entitled 'Ambivalent partnership of the Drosophila posterior class Hox protein Abdominal-B with the Extradenticle and Homothorax cofactors' to PLOS Genetics.

The manuscript was fully evaluated at the editorial level and by independent peer reviewers. The reviewers appreciated the attention to an important topic but identified some concerns that we ask you address in a revised manuscript.

We therefore ask you to modify the manuscript according to the review recommendations. Your revisions should address the specific points made by each reviewer.

To resubmit, log into your Editorial Manager account and select the option 'Revise Submission' in the 'Submissions Needing Revision' folder.

Yours sincerely,

Lolitika Mandal, Ph.D

Academic Editor

PLOS Genetics

Giovanni Bosco

Section Editor

PLOS Genetics

Reviewer's Responses to Questions

**Comments to the Authors:**

Reviewer #1: How proteins have evolved across animal kingdom with diverse functions, most generally specifying segment-specific developmental pathways along with the antero-posterior axis. Interestingly, while Hox proteins themselves have not evolved much, evolution of diversity in body plan in closely related species is attributed to variations in the way Hox proteins interact with their co-factors and/or changes in regulatory/coding sequences of their targets.

This MS by - Curt et al - based on the work done in the laboratories of Graba and Sanchez-Herrero - is an example of how function of AdbdB to suppress the development of 7th abdominal segment. They have shown that in this context, Exd/Hth interact with AbdB differently than in other regions during development. They have shown that expression patterns/levels of Exd/Hth is inversely related to that of AbdB, still the three are required together for their biological function.

The study is well planned, experiments are well executed and results are consistent with their hypothesis and interpretations.

This MS is also an example of fine genetics experiments that authors have carried out.

I recommend its publications in Genetics, while have following comments/suggestions for its further improvement.

1. The work is incomplete without showing how Ex/Hth levels are maintained.

2. What happens when both Hth and Exd are over-expressed or knocked-down?

3. IP experiments suggest Hth can bind to AbdB function independent of Exd. A6 to A5 (more anterior segment) transformations are seen in both over- and loss-of-expression of Hth. Is it possible that when Hth is over-expressed, Exd becomes a limiting factor making hth-AbdB complex dominant negative to Hth/Exd/AbdB complex?

4. The Co-IP experimental results should be quantitated against input levels.

5. Figures legends need to be re-written. For example,

4 (B) Expression of wg-GFP in the A4 segment of males of the following genotypes: wg-GFP/UAS-y+; pnr-Gal4 tub-Gal80ts/UAS-cherry (n=10), wg-GFP/UAS-Abd-B; pnr-Gal4 tub- Gal80ts/UAS-y+ (n=10) and UAS-exd; wg-GFP/UAS-Abd-B; pnr-Gal4 tub-Gal80ts/+ (n=12), all shifted from 18ºC to 29ºC in third larval stage, showing large coincident expression of GFP

55 and cherry in the control, partial repression of wg-GFP when Abd-B is expressed in the pnr domain and partial suppression of this repression if exd is also expressed in this mutant combination.

Very confusing sentence. I guess, authors want to say that when Abd-B is over-expressed, Wg expression is suppressed and this is reversed/rescued when Exd is also over-expressed. This has not come out clearly. All figure legends may be thoroughly edited to make the sentences simpler and clear.

Reviewer #2: This is an interesting and complex paper in which the levels of expression as well as the function of Abd-B in the interaction with Exd-Hth is analysed in Drosophila melanogaster. In the past, two publications in PLoS genetics ( Sambrani et al. 2103; Rivas et al. 2013), demonstrated that, contrary to other Hox genes, the Abd-B Hox protein has an antagonistic relationship with the Hox cofactors Exd-Hth during development. In the current study Curt and collaborators extend this work and explore the interaction of Abd-B with the hox cofactors in further tissues. They show that in different tissues the Abd-B Exd-Hth interaction may be antagonistic while in others the interaction is synergistic as it is accepted for more anterior Hox proteins. Besides the complexity that such tissue/organ differences on Abd-B function reveal, the authors show that the cross regulation that Abd-B can have on Hth-Exd levels of expression and vice versa add an extra level of complexity to any study of this kind. The paper is very thorough, analysing genetic interactions, protein-protein biochemical interactions by Co-IP, protein complex DNA interactions by EMSA and in vivo protein interactions using BiFC analysis. The authors also mutagenize the Abd-B homeodomain and flanking regions to study how different amino acids contribute to the synergistic or antagonistic Abd-B interactions.

The authors conclude that the relationship of Abd-B with the Hox cofactors can be either antagonistic, synergistic or dispensable (probably when the Abd-B protein acts as a monomer or with unknown co-factors) depending on the organ where it functions.

Specific comments:

- The authors should credit in the abstract previous work showing Abd-B Exd-Hth antagonism.

- Except when a direct target is analysed, as is the case for CyclinE or Dll enhancers, the phenotypes obtained are not necessarily due to a single interaction of the Abd-B-Exd-Hth complex on one target but a sum of several. This may confound the interpretations as some of the observed effects can also be exerted independently by Abd-B acting as a monomer or by Exd-Hth controlling Hox independent functions (for example, Engrailed function has been shown to be affected by Exd-Hth and other unknown proteins may also be interacting). This should be discussed.

- The effect on pigmentation caused by ectopic Exd-Hth (Fig 3D and pg27 Discussion) is described as an antagonistic competition with Abd-B. In my opinion this is not that clear. Ectopic Exd-Hth expression also affects pigmentation in segments where Abd-B is not expressed, and the repression of pigmentation in the A4-5 abdominal segments does not agree with lack of anterior Hox expression. The lack of pigmentation could be a Hox independent effect. The formation of ectopic trichomes seems like a more convincing antagonistic phenotype, although the authors should present in Fig 3 close up images that show the trichomes more clearly.

- In the text Pg 17 it says that the Abd-B wild type and Abd-B w protein rescuing ability of A7 suppression is very similar, but in Fig 6E it marks the interaction as highly significant (p<0.0005). Similarly, in pg 18 it says the Tryptophan residue is dispensable for A7 suppression. I believe these incongruences may be due to a mistake on Fig 6E as it seems that the n.s. and **** symbols are displaced one column to the left (and as a result there is a symbol missing over the TG column). If this was not a mistake on the figure, the text should be modified as the difference is highly significant.

- in pg 18 the authors talk about the rescue of the vaginal teeth by rescuing with different UAS-Abd-B variants the lack of genitalia present in Abd-B LDN/Abd-B M1 females. As to form vaginal teeth it is necessary to have a genital disc, it is unclear why the authors insist on talking exclusively about restoring the vaginal teeth function (5th line on page 19) when what they do is to rescue the genital disc. Counting the vaginal teeth is a good way to quantify the genital disc rescue, but there is no reason to say that the proteins are specifically rescuing the vaginal teeth. This also applies to Fig.9 where the antagonism is on genital disc formation and not on vaginal teeth formation. Similarly, in 5th line pg 20, pg 28 last line and pg 30 5th line, it should talk about genitalia formation rather than vaginal teeth development.

- In pg 19-20: The fact that KK mutations and to a lesser extent the TG and CEN variants have increased wg-GFP repressive activity in the wing pouch where Hth is not expressed indicates that there is an unknown factor modulating Abd-B function that in other tissues can or cannot be modulating the Hth-Exd Abd-B interaction.

-Pg 9 “Thus, while wild type expression of Abd-B in A7 (…) in A3-A5 does.”

Why is A6 omitted?

- Pg 12 describe what genes are affected by DpP5. Only Abd-B or more Bx-C elements?

- In pg 17 the CEN mutation is said to be KRKP>CRTQ but in Fig. panel 6B it is shown as GRTQ. Which one is correct?

-Pg 20. The authors should show the outcome of over expressing Exd-Hth in the CNS to conclude there is pure synergism in the CNS. The situation could be similar to what they observed for A7 repression.

- Fig.1F should present the effect of clones or pnr-Gal4 expression in A6-A7 to show if these segments behave differently.

Minor comments:

- Drosophila in italics all over the text.

-Pg 9. If at later stages the Hth increase is seen both in experimental and control experiments (Sup. Fig.3), is this relevant for the paper? I would consider deleting the sentence starting with: “At later stages (about 34-38h…)“.

- Pg 20: Modify title. Mutations are not required. Especially in this case. It is the W a.a. that is required for Abd-B CNS function.

- Lettering in Fig. 8 has to be improved, the resolution is not good.

-Fig 6 legend: “…most conserved domains…” specify where: in insects, arthropods or also in vertebrates?

- Pg. 25. “Consistently, ectopic Abd-B (…) exd expression.” Does this refer to the wing, the histoblasts or both?

- Pg. 25. “This highlights the importance (…) proposed for Ubx …” An earlier paper should also be cited: Irvine et al. 1993 Development 117, 387-99.

- Pg 31 “as about 25% longer”. Shouldn’t it say shorter or faster?

Reviewer #3: This manuscript, titled “Ambivalent partnership of the Drosophila posterior class Hox protein Abdominal-B with the Extradenticle and Homothorax cofactors “ by Curt, et al defines interactions between Abd-B, Hth and Exd in different contexts in Drosophila development. A careful examination of the wild-type expression of exc, hth and Abd-B in A7 is followed by a close examination of cross-regulation by these factors on one another. Both LOF and GOF of hth/exd were shown to lead to anterior transformations of A6 in males and mis-regulation of Wg. CoIP experiments examine interactions between these proteins in various contexts. Finally, specific point mutations in Abd-B in the YWPM-anologous region and C-terminal of the homeodomain shows context-dependent activities of many residues on the phenotype in several tissues in which Abd-B plays a role. This study has been meticulously conducted, is presented well overall and this reviewer applauds the breadth and depth of the study, genetically and mechanistically. It adds significantly to our understanding of AbdB Hox function with Exd and Hth in Drosophila. It also brings new insight to the mechanisms potentially used by all posterior Hox genes in other organisms.

One suggestion for author consideration: the discussion is nearly as complex as the results. There may be an opportunity to do less recapitulating of the results and more discussion of implications (which is already there but might be expanded). This reviewer, for instance, is intrigued by the idea that the context-dependent activities could be related to specific expression of Hox, Exd or Hth as well as other co-factors in tissues/organs as they develop past segments, an idea supported by some of the data presented.

Also for author consideration: Would “Context-dependent” be better word choice in title than “Ambivalent” for title – seems anthropomorphic to this reviewer? And perhaps drop the word ‘cofactors’ as, at least in some of their roles, it seems they aren’t acting as cofactors?

Minor comments:

Page 8: for non-Drosophilists, can the authors more carefully describe ‘extruded’ for the 7th abdominal segment. It is formed but lost from pupae?

Likewise, Figure 1 and 2 are presented in such a way that a non-Drosophila geneticist can pretty easily understand the genetics shown. This is more difficult in many following figures. The authors could consider providing a bit of additional information in the figures to help with this aspect.

**Have all data underlying the figures and results presented in the manuscript been provided?**

Reviewer #1: Yes

Reviewer #2: Yes

Reviewer #3: Yes

PLOS authors have the option to publish the peer review history of their article (what does this mean?). If published, this will include your full peer review and any attached files.

Reviewer #1: No

Reviewer #2: No

Reviewer #3: No

---

## [Decision Letter · Decision Letter 1]

9 Dec 2024

Dear Dr Sánchez-Herrero,

We are pleased to inform you that your manuscript entitled "Ambivalent partnership of the Drosophila posterior class Hox protein Abdominal-B with Extradenticle and Homothorax" has been editorially accepted for publication in PLOS Genetics. Congratulations!

Yours sincerely,

Lolitika Mandal, Ph.D

Academic Editor

PLOS Genetics

Giovanni Bosco

Section Editor

PLOS Genetics

Aimée Dudley

Editor-in-Chief

PLOS Genetics

Anne Goriely

Editor-in-Chief

PLOS Genetics

Comments from the reviewers (if applicable):

Reviewer's Responses to Questions

**Comments to the Authors:**

Reviewer #1: I am happy with the response and the revision.

Reviewer #2: The authors have answered my querries

Reviewer #3: Revisions have addressed concerns raised by this reviewer.

**Have all data underlying the figures and results presented in the manuscript been provided?**

Reviewer #1: None

Reviewer #2: Yes

Reviewer #3: Yes

PLOS authors have the option to publish the peer review history of their article (what does this mean?). If published, this will include your full peer review and any attached files.

Reviewer #1: No

Reviewer #2: No

Reviewer #3: No

**Data Deposition**

http://datadryad.org/submit?journalID=pgenetics&manu=PGENETICS-D-24-00718R1

**Press Queries**

---

## [Editor Report · Acceptance letter]

6 Jan 2025

PGENETICS-D-24-00718R1 

Ambivalent partnership of the Drosophila posterior class Hox protein Abdominal-B with Extradenticle and Homothorax 

Dear Dr Sánchez-Herrero, 

We are pleased to inform you that your manuscript entitled "Ambivalent partnership of the Drosophila posterior class Hox protein Abdominal-B with Extradenticle and Homothorax" has been formally accepted for publication in PLOS Genetics! Your manuscript is now with our production department and you will be notified of the publication date in due course.

With kind regards,

Anita Estes

PLOS Genetics

On behalf of:
